# Optimizing Functionals on the Space of Probabilities with Input Convex Neural Networks

**David Alvarez-Melis**                                                  *daalvare@microsoft.com*
*Microsoft Research*

**Yair Schiff**                                                          *yair.schiff@ibm.com*
*IBM Watson*

**Youssef Mroueh**                                                       *mroueh@us.ibm.com*
*IBM Research AI*

**Reviewed on OpenReview:** *https://openreview.net/forum?id=dpOYN7o8Jm*

## Abstract

Gradient flows are a powerful tool for optimizing functionals in general metric spaces, including the space of probabilities endowed with the Wasserstein metric. A typical approach to solving this optimization problem relies on its connection to the dynamic formulation of optimal transport and the celebrated Jordan-Kinderlehrer-Otto (JKO) scheme. However, this formulation involves optimization over convex functions, which is challenging, especially in high dimensions. In this work, we propose an approach that relies on the recently introduced input-convex neural networks (ICNN) to parametrize the space of convex functions in order to approximate the JKO scheme, as well as in designing functionals over measures that enjoy convergence guarantees. We derive a computationally efficient implementation of this JKO-ICNN framework and experimentally demonstrate its feasibility and validity in approximating solutions of low-dimensional partial differential equations with known solutions. We also demonstrate its viability in high-dimensional applications through an experiment in controlled generation for molecular discovery.

## 1 Introduction

Numerous problems in machine learning and statistics can be formulated as finding a probability distribution that minimizes some objective function of interest. One recent example of this formulation is generative modeling, where one seeks to model a data-generating distribution $\rho_{\text{data}}$ by finding, among a parametric family $\rho_\theta$, the distribution that minimizes some notion of discrepancy to $\rho_{\text{data}}$, i.e., $\min_\theta D(\rho_\theta, \rho_{\text{data}})$. Different choices of discrepancies give rise to various training paradigms, such as generative adversarial networks (Goodfellow et al., 2014) (Jensen-Shannon divergence), Wasserstein GAN (Arjovsky et al., 2017) (1-Wasserstein distance) and maximum likelihood estimation (KL divergence) (Murphy, 2012; Rezende & Mohamed, 2015; Kingma & Welling, 2013). In general, such problems can be cast as finding $\rho^* = \operatorname{argmin}_\rho F(\rho)$, for a functional $F$ on distributions.

Beyond machine learning and statistics, optimization on the space of probability distributions is prominent in applied mathematics, particularly in the study of partial differential equations (PDE). The seminal work of Jordan et al. (1998), and later Otto (2001), Ambrosio et al. (2005), and several others, showed that many classic PDEs can be understood as minimizing certain functionals defined on distributions. Central to these works is the notion of *gradient flows* on the probability space endowed with the Wasserstein metric. Jordan, Kinderlehrer, and Otto (1998) set the foundations of a theory establishing connections between optimal transport, gradient flows, and differential equations. In addition, they proposed a general iterative method, popularly referred to as the JKO scheme, to solve PDEs of the Fokker-Planck type. This method was later extended to more general PDEs and in turn to more general functionals over probability space (Ambrosio et al., 2005). The JKO scheme can be seen as a generalization of the implicit Euler method on the

probability space endowed with the Wasserstein metric. This approach has various appealing theoretical convergence properties owing to a notion of convexity of probability functionals, known as geodesic convexity (see Santambrogio (2017) for more details).

Several computational approaches to JKO have been proposed, among them an elegant method introduced in Benamou et al. (2014) that reformulates the JKO variational problem on probability measures as an optimization problem on the space of convex functions. This reformulation is made possible thanks to Brenier's Theorem (Brenier, 1991). However, the appeal of this computational scheme comes at a price: computing updates involves solving an optimization over convex functions at each step, which is challenging in general. The practical implementations in Benamou et al. (2014) make use of space discretization to solve this optimization problem, which limits their applicability beyond two dimensions.

In this work, we propose a computational approach to the JKO scheme that is scalable in high-dimensions. At the core of our approach are Input-Convex Neural Networks (ICNN) (Amos et al., 2017), a recently proposed class of deep models that are convex with respect to their inputs. We use ICNNs to find parametric solutions to the reformulation of the JKO problem as optimization on the space of convex functions by Benamou et al. (2014). This leads to an approximation of the JKO scheme that we call JKO-ICNN. In practice, we implement JKO-ICNN with finite samples from distributions and optimize the parameters of ICNNs with adaptive gradient descent using automatic differentiation.

To evaluate the soundness of our approach, we first conduct experiments on well-known PDEs in low dimensions that have exact analytic solutions, allowing us to quantify the approximation quality of the gradient flows evolved with our method. We then use our approach in a high-dimensional setting, where we optimize a dataset of molecules to satisfy certain properties, such as drug-likeness (QED). The results show that our JKO-ICNN approach is successful at approximating solutions of PDEs and has the unique advantage of scalability in terms of optimizing generic probability functionals on the probability space in high dimensions. When compared to direct optimization methods or particle gradient flows methods, JKO-ICNN has the advantage of computational stability and amortization of computational cost, since the maps found while training JKO-ICNN on one sample from a distribution $\rho_0$ generalize at transporting a new sample unseen during the training at no additional cost.

## 2 Background

**Notation** Let $\mathcal{X}$ be a Polish space equipped with metric $d$ and $\mathcal{P}(\mathcal{X})$ be the set of non-negative Borel measures with finite second-order moment on that space. The space $\mathcal{P}(\mathcal{X})$ contains both continuous and discrete measures, the latter represented as an empirical distribution: $\sum_{i=1}^{N} p_i \delta_{x_i}$, where $\delta_x$ is a Dirac at position $x \in \mathcal{X}$. For a measure $\rho$ and measurable map $T : \mathcal{X} \to \mathcal{X}$, we use $T_{\sharp}\rho$ to denote the push-forward measure, and $\mathbf{J}_T$ the Jacobian of $T$. For a function $u : \mathcal{X} \to \mathbb{R}$, $\nabla u$ is the gradient and $\mathbf{H}_u(x)$ is the Hessian. $\nabla \cdot$ denotes the divergence operator. For a matrix $A$, $|A|$ denotes its determinant. When clear from the context, we use $\rho$ interchangeably to denote a measure and its density.

**Gradient flows in Wasserstein space** Consider first a functional $F : \mathcal{X} \to \mathbb{R}$ and a point $x_0 \in \mathcal{X}$. A *gradient flow* is an absolutely continuous curve $x(t)$ that evolves from $x_0$ in the direction of steepest descent of $F$. When $\mathcal{X}$ is Hilbertian and $F$ is sufficiently smooth, its gradient flow can be expressed as the solution of a differential equation $x'(t) = -\nabla F(x(t))$, with initial condition $x(0) = x_0$.

Gradient flows can be defined in probability space too, as long as a suitable notion of distance between probability distributions is chosen. Formally, let us consider $\mathcal{P}(\mathcal{X})$ equipped with the $p$-Wasserstein distance, which for measures $\alpha, \beta \in \mathcal{P}(\mathcal{X})$ is defined as:

$$\mathrm{W}_p(\alpha, \beta) \triangleq \min_{\pi \in \Pi(\alpha, \beta)} \int \|x - y\|_2^2 \, \mathrm{d}\pi(x, y). \tag{1}$$

Here $\Pi(\alpha, \beta)$ is the set of couplings (*transportation plans*) between $\alpha$ and $\beta$, formally: $\Pi(\alpha, \beta) \triangleq \{\pi \in \mathcal{P}(\mathcal{X} \times \mathcal{X}) \mid P_{1\sharp}\pi = \alpha, P_{2\sharp}\pi = \beta\}$. Endowed with this metric, the *Wasserstein space* $\mathbb{W}_p(\mathcal{X}) = (\mathcal{P}(\mathcal{X}), \mathrm{W}_p)$ is a complete and separable metric space. In this case, given a functional in probability space $F : \mathcal{P}(\mathcal{X}) \to \mathbb{R}$, its gradient flow in $\mathbb{W}_p(\mathcal{X})$ is a curve $\rho(t) : \mathbb{R}_+ \to \mathcal{P}(\mathcal{X})$ that satisfies $\partial_t \rho(t) = -\nabla_{\mathbb{W}_2} F(\rho(t))$. Here $\nabla_{\mathbb{W}_2}$ is a

natural notion of gradient in $\mathbb{W}_p(\mathcal{X})$ given by: $\nabla_{\mathbb{W}_2} F(\rho) = -\nabla \cdot \left( \rho \nabla \frac{\delta F}{\delta \rho} \right)$ where $\frac{\delta F}{\delta \rho}$ is the first variation of the functional $F$. Therefore, the gradient flow of $F$ solves the following PDE:

$$\partial_t \rho(t) - \nabla \cdot \left( \rho(t) \nabla \left( \frac{\delta F}{\delta \rho}(\rho(t)) \right) \right) = 0 \tag{2}$$

also known as a continuity equation.

## 3 Gradient flows via JKO-ICNN

In this section we introduce the JKO scheme for solving gradient flows, show how to cast it as optimization over convex functions, and propose a method to solve the resulting problem via ICNN parametrization.

### 3.1 JKO scheme on measures

Throughout this work we consider problems of the form $\min_{\rho \in \mathcal{P}(\mathcal{X})} F(\rho)$, where $F : \mathcal{P}(\mathcal{X}) \to \mathbb{R}$ is a functional over probability measures encoding some objective of interest. Following the gradient flow literature (e.g., Santambrogio (2015; 2017)), we focus on three quite general families of functionals:

$$\mathcal{F}(\rho) = \int f(\rho(x)) \, \mathrm{d}x, \qquad \mathcal{V}(\rho) = \int V(x) \, \mathrm{d}\rho, \qquad \mathcal{W}(\rho) = \frac{1}{2} \iint W(x - x') \, \mathrm{d}\rho(x) \, \mathrm{d}\rho(x'), \tag{3}$$

where $f : \mathbb{R} \to \mathbb{R}$ is convex and superlinear and $V, W : \mathcal{X} \to \mathbb{R}$ are convex and sufficiently smooth. These functionals are appealing for various reasons. First, their gradient flows enjoy desirable convergence properties, as we discuss below. Second, they have a physical interpretation as internal, potential, and interaction energies, respectively. Finally, their corresponding continuity equations (2) turn out to recover various classic PDEs (see Table 1 for equivalences). Thus, in this work, we focus on objectives that can be written as linear combinations of these three types of functionals.

**Table 1: Equivalence between gradient flows and PDEs.** In each case, the gradient flow of the functional $F(\rho)$ in Wasserstein space in the rightmost column satisfies the PDE in the middle column.

| Class | PDE $\partial_t \rho =$ | Flow Functional $F(\rho) =$ |
|---|---|---|
| Heat Equation | $\Delta \rho$ | $\int \rho(x) \log \rho(x) \, \mathrm{d}x$ |
| Advection | $\nabla \cdot (\rho \nabla V)$ | $\int V(x) \, \mathrm{d}\rho(x)$ |
| Fokker-Planck | $\Delta \rho + \nabla \cdot (\rho \nabla V)$ | $\int \rho(x) \log \rho(x) \, \mathrm{d}x + \int V(x) \, \mathrm{d}\rho(x)$ |
| Porous Media | $\Delta(\rho^m) + \nabla \cdot (\rho \nabla V)$ | $\frac{1}{m-1} \int \rho(x)^m \, \mathrm{d}x + \int V(x) \, \mathrm{d}\rho(x)$ |
| Adv.+Diff.+Inter. | $\nabla \cdot \left[ \rho(\nabla f'(\rho) + \nabla V + (\nabla W) * \rho) \right]$ | $\int V(x) \, \mathrm{d}\rho(x) + \int f(\rho(x)) \, \mathrm{d}x + \frac{1}{2} \iint W(x - x') \, \mathrm{d}\rho(x) \, \mathrm{d}\rho(x')$ |

For a functional $F$ of this form, it can be shown that the corresponding gradient flow defined in Section 2 converges exponentially fast to a unique minimizer (Santambrogio, 2017). This suggests solving the optimization problem $\min_\rho F(\rho)$ by following the gradient flow, starting from some initial configuration $\rho_0$. A convenient method to study this PDE is through the time discretization provided by the Jordan–Kinderlehrer–Otto (JKO) iterated movement minimization scheme (Jordan et al., 1998):

$$\rho_{t+1}^\tau \in \operatorname*{argmin}_{\rho \in \mathbb{W}_2(\mathcal{X})} F(\rho) + \frac{1}{2\tau} \mathrm{W}_2^2(\rho, \rho_t^\tau), \tag{4}$$

where $\tau > 0$ is a time step parameter. This scheme will form the backbone of our approach.

### 3.2 From measures to convex functions

The general JKO scheme (4) discretizes the gradient flow (and therefore, the corresponding PDE) in time, but it is still formulated on —potentially infinite-dimensional, and therefore intractable— probability space $\mathcal{P}(\mathcal{X})$. Obtaining an implementable algorithm requires recasting this optimization problem in terms of a space that is easier to handle than that of probability measures. As a first step, we do so using convex functions.

A cornerstone of optimal transport theory states that for absolutely continuous measures and suitable cost functions, the solution of the Kantorovich problem concentrates around a deterministic map $T$ (the

Monge map). Furthermore, for the quadratic cost, Brenier's theorem (Brenier, 1991) states that this map is given by the gradient of a convex function $u$, i.e., $T(x) = \nabla u(x)$. Hence given a measure $\alpha$, the mapping $u \in \text{cvx}(\mathcal{X}) \mapsto (\nabla u)_\sharp \alpha \in \mathcal{P}(\mathcal{X})$ can be seen as a parametrization, which depends on $\alpha$, of the space of probabilities (McCann, 1997). We furthermore have for any $u \in \text{cvx}(\mathcal{X})$:

$$W_2^2(\alpha, (\nabla u)_\sharp \alpha) = \int_{\mathcal{X}} \|\nabla u(x) - x\|_2^2 \, d\alpha. \tag{5}$$

Using this expression and the parametrization $\rho = (\nabla u)_\sharp \rho_t^\tau$ in Problem (4), we obtain a reformulation of Wasserstein gradient flows as optimization over convex functions (Benamou et al., 2014):

$$u_{t+1}^\tau \in \underset{u \in \text{cvx}(\mathcal{X})}{\text{argmin}} \, F((\nabla u)_\sharp \rho_t^\tau) + \tfrac{1}{2\tau} \int_{\mathcal{X}} \|\nabla u(x) - x\|_2^2 \, d\rho_t^\tau, \tag{6}$$

which implicitly defines a sequence of measures via $\rho_{t+1}^\tau = (\nabla u_{t+1}^\tau)_\# (\rho_t^\tau)$. For potential and interaction functionals, Lemma 3.1 shows that the first term in this scheme can be written in a form amenable to optimization on $u$.

**Lemma 3.1** (Potential and Interaction Energies). *For the pushforward measure $\rho = (\nabla u)_\sharp \rho_t$, the functionals $\mathcal{V}$ and $\mathcal{W}$ can be written as:*

$$\begin{aligned} \mathcal{V}(\rho) &= \int (V \circ \nabla u)(x) \, d\rho_t(x) \\ \mathcal{W}(\rho) &= \frac{1}{2} \iint W(\nabla u(x) - \nabla u(y)) \, d\rho_t(y) \, d\rho_t(x). \end{aligned} \tag{7}$$

Crucially, $\rho_t$ appears here only as the integrating measure. We will exploit this property for finite-sample computation in the next section. In the case of internal energies $\mathcal{F}$, however, the integrand itself depends on $\rho_t$, which poses difficulties for computation. To address this, we start in Lemma 3.2 by tackling the change of density when using strictly convex potential pushforward maps:

**Lemma 3.2** (Change of variable). *Given a strictly convex $u \in \text{cvx}(\mathcal{X})$, $\nabla u$ is invertible, and $(\nabla u)^{-1} = \nabla u^*$, where $u^*$ is the convex conjugate of $u$, $u^*(y) = \sup_{x \in dom(u)} \langle x, y \rangle - u(x)$. Given a measure $\alpha$ with density $\rho_\alpha$, the density $\rho_\beta$ of the measure $\beta = (\nabla u)_\sharp \alpha$ is given by:*

$$\rho_\beta(y) = \frac{\rho_\alpha}{|\mathbf{H}_u|} \circ (\nabla u)^{-1}(y) = \frac{\rho_\alpha}{|\mathbf{H}_u|} \circ \nabla u^*(y).$$

In other words $\log\big(\rho_\beta(y)\big) = \log\big(\rho_\alpha(\nabla u^*(y))\big) - \log(|\mathbf{H}_u(\nabla u^*(y))|)$. Iterating Lemma 3.2 across time in the JKO steps we obtain:

**Corollary 3.3** (Iterated Change of Variables in JKO). *Assume $\rho_0$ has a density. Let $T_{1:t}^\tau = \nabla u_t^\tau \cdots \circ \nabla u_1^\tau$, where $u_t^\tau$ are optimal convex potentials in the JKO sequence that we assume are strictly convex. We use the convention $T_{1:0}^\tau(x) = x$. We have $(T_{1:t}^\tau)^{-1} = \nabla(u_1^\tau)^* \circ \cdots \circ \nabla(u_t^\tau)^*$ where $(u_t^\tau)^*$ is the convex conjugate of $u_t^\tau$. At time $t$ of the JKO iterations we have: $\rho_t(x) = T_{1:t}^\tau \rho_0(x)$, and therefore:*

$$\log(\rho_t^\tau) = \left( \log(\rho_0) - \sum_{s=1}^{t} \log(|\mathbf{H}_{u_s^\tau}(T_{1:s-1}^\tau)|) \right) \circ (T_{1:t}^\tau)^{-1}.$$

From Corollary 3.3, we see that the iterates in the JKO scheme imply a change of densities that shares similarities with normalizing flows (Rezende & Mohamed, 2015; Huang et al., 2021), where the depth of the flow network corresponds to the time in JKO. Whereas the normalizing flows of Rezende & Mohamed (2015) draw connections to the Fokker-Planck equation in the generative modeling context, JKO is more general and allows for rigorous optimization of generic functionals on the probability space.

Armed with this expression of $\rho_t^\tau$, we can now write $\mathcal{F}$ in terms of the convex potential $u$:

**Lemma 3.4** (Internal Energy). *Let $\rho_t$ be the measure at time $t$ of the JKO iterations. In the notation of Corollary 3.3, for the measure $\rho = (\nabla u)_\sharp \rho_t = (\nabla u \circ T_{1:t})_\sharp \rho_0$, we have:*

$$\mathcal{F}(\rho) = \int f(\xi(x))|\mathbf{H}_u(T_{1:t}(x))|\,|\mathbf{J}_{T_{1:t}}(x)|\,\mathrm{d}x, \tag{8}$$

*where $\xi(x) = \rho_0(x)/(|\mathbf{H}_u(T_{1:t}(x))||\mathbf{J}_{T_{1:t}}(x)|)$. Assuming $\rho_0 > 0$, we have: $\mathcal{F}(\rho) = \mathbb{E}_{x \sim \rho_0} f \circ \xi(x)/\xi(x)$.*

### 3.3 From convex functions to finite parameters

Solving Problem (6) requires: (i) a tractable parametrization of $\mathrm{cvx}(\mathcal{X})$, the space of convex functions, (ii) a method to evaluate and compute gradients of the Wasserstein distance term, and (iii) a method to evaluate and compute gradients of the functionals as expressed in Lemmas 3.1 and 3.4.

For (i), we rely on the recently proposed Input Convex Neural Networks (Amos et al., 2017). See Appendix B.1 for a background on ICNN. Given $\rho_0 = \frac{1}{n}\sum_{i=1}^{n}\delta_{x_i}$ we solve for $t = 1\ldots T$:

$$\theta_{t+1}^\tau \in \underset{\theta:\, u_\theta \in \mathrm{ICNN}(\mathcal{X})}{\mathrm{argmin}} \quad L(\theta), \tag{9}$$

$$L(\theta) \triangleq F((\nabla_x u_\theta(x))_\sharp \rho_t^\tau) + \frac{1}{2\tau}\int_{\mathcal{X}} \|\nabla_x u_\theta(x) - x\|_2^2 \,\mathrm{d}\rho_t^\tau$$

where $\mathrm{ICNN}(\mathcal{X})$ is the space of Input Convex Neural Networks, and the $\theta$ denotes parameters of the ICNN. Problem (9) defines a JKO sequence of measures via $\rho_{t+1}^\tau = (\nabla_x u_{\theta_{t+1}^\tau})_\#(\rho_t^\tau)$. We call this iterative process JKO-ICNN, where each optimization problem can be solved with gradient descent on the parameter space of the ICNN, using backpropagation and automatic differentiation.

For (ii), we note that this term can be interpreted as an expectation, namely, $\frac{1}{2\tau}\mathbb{E}_{x \sim \rho_t^\tau}\|\nabla_x u_\theta(x) - x\|_2^2$, so we can approximate it with finite samples (*particles*) of $\rho_t^\tau$ obtained via the pushforward map of previous point clouds in the JKO sequence, i.e., using $\frac{1}{2\tau n}\sum_{i=1}^{n}\|\nabla_x u_\theta(x_i) - x_i\|_2^2$, $\{x_i\}_{i=1}^{n} \sim \rho_t^\tau$. Finally, for (iii) we first note that the $\mathcal{V}$ and $\mathcal{W}$ functionals can also be written as expectations over $\rho_t^\tau$:

$$\mathcal{V}\big((\nabla_x u_\theta)_\sharp \rho_t^\tau\big) = \underset{x \sim \rho_t^\tau}{\mathbb{E}} V(\nabla_x u_\theta(x)), \tag{10}$$

$$\mathcal{W}\big((\nabla_x u_\theta)_\sharp \rho_t^\tau\big) = \tfrac{1}{2}\underset{x,y \sim \rho_t^\tau}{\mathbb{E}} W(\nabla_x u_\theta(x) - \nabla_x u_\theta(y)).$$

Thus, as long as we can parametrize the functions $V$ and $W$ in a differentiable manner, we can estimate the value and gradients of these two functionals through finite samples too. In many cases $V$ and $W$ will be simple analytic functions, such as in the PDEs considered in Section 5. To model more complex optimization objectives with these functionals we can leverage ICNNs once more to parametrize the functions $V$ and $W$ as neural networks in a way that enforces their convexity. This is what we do in the molecular discovery experiments in Section 6.

**Particular cases of internal energies** Equation (8) simplifies for some choices of $f$, e.g., for $f(t) = t\log t$ (which yields the heat equation) and strictly convex $u$ we get:

$$\mathcal{F}\big((\nabla_x u_\theta)_\sharp \rho_t\big) = \int \frac{\rho_t(x)}{|\mathbf{H}_{u_\theta}(x)|}\log\frac{\rho_t(x)}{|\mathbf{H}_{u_\theta}(x)|}|\mathbf{H}_{u_\theta}(x)|\,\mathrm{d}x = \mathcal{F}(\rho_t) - \int \log|\mathbf{H}_{u_\theta}(x)|\rho_t(x)\,\mathrm{d}x \tag{11}$$

where we drop $\tau$ from the notation for simplicity. This expression has a notable interpretation: pushing forward measure $\rho_t$ by $\nabla u_\theta$ increases its entropy by a log-determinant barrier term on $u_\theta$'s Hessian. Note that only the second term in equation (11) depends on $u_\theta$, hence $\nabla_\theta \mathcal{F}\big((\nabla_x u_\theta)_\sharp \rho_t\big) = -\nabla_\theta \mathbb{E}_{x \sim \rho_t}\log|\mathbf{H}_{u_\theta}(x)|$. Since the latter can be approximated—as before— by an empirical expectation, it can be used as a surrogate objective for optimization.

Another notable case is given by $f(t) = \frac{1}{m-1}t^m, m > 1$, which yields a nonlinear diffusion term as in the porous medium equation (Table 1). In this case, equation (8) becomes $\mathcal{F}(\rho) = \frac{1}{m-1}\mathbb{E}_{x \sim \rho_0}\xi(x)^{m-1}$, whose gradient with respect to $\theta$ is:

$$-\underset{x \sim \rho_0}{\mathbb{E}}\exp\{(m-1)\log\xi(x)\}\nabla_\theta\log|\mathbf{H}_{u_\theta}(T_{1:t}(x))| \tag{12}$$

Table 3 in the Appendix collects all the surrogate objectives described in this section.

---

**Algorithm 1:** JKO-ICNN: JKO variational scheme using input convex neural networks

---

**Inputs:**
- Functional to optimize: $F(\rho)$
- Outer loop (JKO scheme) parameters: step size $\tau > 0$ and number of steps $T \in \mathbb{N}$
- Inner loop (ICNN fitting) parameters: learning rate $\eta > 0$ and number of iterations $n_u \in \mathbb{N}$
- *warmstart* boolean: whether to initialize ICNN weights from prior iteration's solution

**Initialize** $u_\theta \in$ ICNN, $\theta$ parameters of ICNN, $\rho_0^\tau = \frac{1}{N} \sum_{i=1}^N \delta_{x_i}$.

**for** $t = 0$ **to** $T - 1$ **do**
  **if** not *warmstart*: $\theta \leftarrow$ InitializeWeights()
  **for** $i = 1$ **to** $n_u$ **do**
    {JKO inner loop: Updating ICNN }
    $L(\theta) = F((\nabla_x u_\theta)_\# \rho_t^\tau) + \frac{1}{2\tau} \mathbb{E}_{\rho_t^\tau} ||x - \nabla_x u_\theta(x)||^2$
    $\theta \leftarrow \text{Adam}(\theta, \eta, \nabla_\theta L_\theta)$
  **end for**
  {JKO outer loop: Updating point cloud (measures) }
  $\rho_{t+1}^\tau = \nabla_x(u_\theta)_\# \rho_t^\tau$
**end for**
**Output:** $\rho_T^\tau$

---

**Implementation and practical considerations**  We implement Algorithm 1 in PyTorch (Paszke et al., 2019), relying on automatic differentiation to solve the inner optimization loop. The finite-sample approximations of $\mathcal{V}$ and $\mathcal{W}$ (equation (10)) can be used directly, but the computation of the surrogate objectives for internal energies $\mathcal{F}$ (equations (11) and (12)) require computing Hessian log-determinants—prohibitive in high dimensions. Thus, we use a stochastic log-trace estimator based on the Hutchinson method (Hutchinson, 1989), as used by Huang et al. (2021) (see Appendix B.4 for details on implementation and complexity of this method). To enforce strong convexity on the ICNN, we clip its weights away from 0 after each update. When needed (e.g., for evaluation), we estimate the true internal energy functional $\mathcal{F}$ by using Corollary 3.3 to compute densities. Appendix B.5 provides full implementation details.

**Remark 3.5** (Approximation of Brenier Potential). *As pointed in Benamou et al. (2014), the Brenier potential can be non smooth and not strictly convex. Thus, JKO-ICNN can be understood as effectively optimizing not on the full space of convex functions, but rather on a smooth subset of it (if we use a smooth activation). As a consequence, JKO-ICNN does not seek 'the' Brenier potential, but instead a family of smooth Brenier potentials, which may be distinct from the former. Note that this argument is similar to the one used in (Paty et al., 2020).*

## 4 Related Work

**Optimization over Probability Measures and Connections to Convex Optimization**  The general problem of optimization over the space of measures has been studied extensively, particularly in the functional analysis and applied mathematics literature (Jordan et al., 1998; Carlen & Gangbo, 2003; Léonard, 2008). Despite its unique challenges, optimization over measures exhibits various similarities to classic (finite-dimensional) optimization. For example, Chizat (2021) recently studied the convergence of Bregman proximal gradient methods for convex objectives over measures — extensions of their classical counterparts (Nesterov, 2004). Similarly, Wibisono (2018) studies sampling as an optimization over measures, revealing deep connections between these two problems.

**Computational gradient flows**  A common general approach for solving optimization problems involving distributions relies on gradient flows, which have been implemented through various computational methods. Benamou et al. (2016) propose an augmented Lagrangian approach for convex functionals based on the dynamical optimal transport implementation of Benamou & Brenier (2000). Another approach relying on the dynamic formulation of JKO and an Eulerian discretization of measures (i.e. via histograms) is the recent primal dual algorithm of Carrillo et al. (2021). Closer to our work is the formulation of Benamou

et al. (2014) that casts the problem as an optimization over convex functions. This work relies on a Lagrangian discretization of measures, via cloud points, and on a representation of convex functions and their corresponding subgradients via their evaluation at these points. This method does not scale well in high dimensions since it computes Laguerre cells in order to find the subgradients. A different approach by Peyré (2015) defines entropic gradient flows using Eulerian discretization of measures and Sinkhorn-like algorithms that leverage an entropic regularization of the Wasserstein distance. Caluya & Halder (2019) propose a method that combines entropic regularization and particle methods to solve gradient flows for the Fokker-Planck equation. Frogner & Poggio (2020) propose kernel approximations to compute gradient flows. On a different line of work, Salim et al. (2020) propose a forward-backward discretization scheme for Wasserstein gradient flows, and show that it enjoys convergences guarantees akin to those of proximal gradient methods in usual Euclidean spaces. Finally, blob methods have been considered in Craig & Bertozzi (2014) and Carrillo et al. (2019) for the aggregation and diffusion equations. Blob methods regularize velocity fields with mollifiers (convolution with a kernel) and –unlike many other alternative methods– allow for the approximation of internal energies.

**Forward and particle descent methods** In addition, particle descent methods were explored for defining gradient flows for various geometries, e.g., for Sliced Wasserstein distance (Bonet et al., 2021; Liutkus et al., 2019), MMD (Mroueh et al., 2019; Mroueh & Rigotti, 2020; Mroueh & Nguyen, 2021; Arbel et al., 2019), Stein discrepancy (Liu & Wang, 2016; Liu, 2017; Duncan et al., 2019; Korba et al., 2021), and Kullback–Leibler divergence (Glaser et al., 2021; Feng et al., 2021; di Langosco et al., 2021). Wang et al. (2021) proposes a projected Wasserstein gradient descent that exploits the manifold assumption.

**ICNN, optimal transport, and generative modeling** ICNN architectures were originally proposed by Amos et al. (2017) to allow for efficient inference in settings like structured prediction, data imputation, and reinforcement learning. Since their introduction, they have been exploited in various other settings that require parametrizing convex functions, including optimal transport. For example, Makkuva et al. (2020) propose using them to learn an explicit optimal transport map between distributions, which under suitable assumptions, can be shown to be the gradient of a convex function (Brenier, 1991). The ICNN parametrization has been also exploited in order to learn continuous Wasserstein barycenters by Korotin et al. (2021). Using this same characterization, Huang et al. (2021) recently proposed to use ICNNs to parametrize flow-based invertible probabilistic models, an approach they call *convex potential flows.* These are instances of *normalizing flows* (not to be confused with *gradient flows*), and are useful for learning generative models when samples from the target (i.e., optimal) distributions are available and the goal is to learn a parametric generative model. Our use of ICNNs differs from these prior works and other approaches to generative modeling in two important ways. First, unlike those, our approach can be used even if the target distribution is available only up to a constant, or even if it cannot be sampled from and is only implicitly characterized as the minimizer of an optimization problem over distributions. Indeed, any functional defined over distributions can be used as a plug-in objective in our framework, making it far more general than traditional generative modeling. Additionally, compared to other works that use ICNNs for optimal transport, we leverage these networks not only for solving a single optimal transport problem, but rather for a sequence of JKO step optimization problems that involve various terms in addition to the Wasserstein distance.

**Neural networks methods for JKO** In concurrent work, Mokrov et al. (2021) and Bunne et al. (2021) adopt a similar approach for approximating JKO with ICNNs. While the former is concerned exclusively with the Fokker-Planck equation, here we consider additional classes of PDEs. Bunne et al. (2021) tackle a different problem: *learning* dynamics with JKO, i.e., the inverse problem of learning the functional whose JKO flow follows empirical observations, which they also approach in its stochastic version using an SDE approach (Bunne et al., 2022). Other subsequent works have built upon these initial set of parallel works. For example, Hwang et al. (2021) provide non-asymptotic approximation guarantees and an empirical validation for a method similar to the one proposed here. Nodozi & Halder (2022), on the other hand, introduce a variant of the JKO-ICNN approach that relies on the alternating direction method of multipliers (ADMM). On a different line of work, a min-max formulation of the JKO scheme was explored by Fan et al. (2021) and a link between proximal training methods of GANs with Wasserstein gradient flows was proposed by Lin et al. (2021).

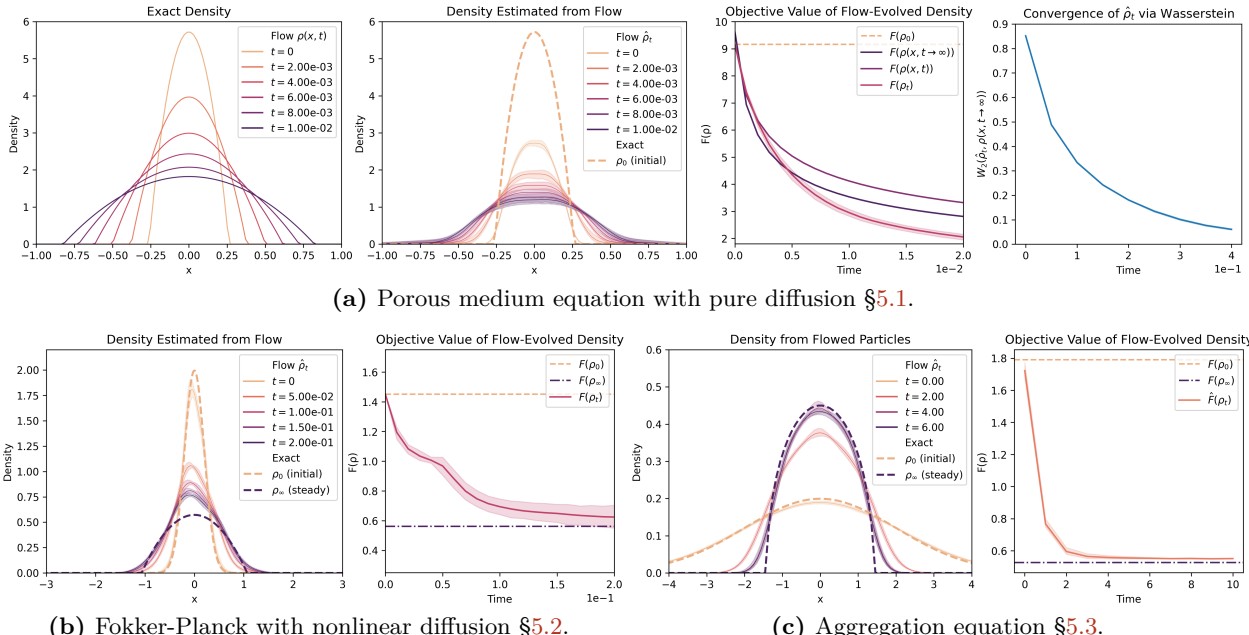

**(a)** Porous medium equation with pure diffusion §5.1.

**(b)** Fokker-Planck with nonlinear diffusion §5.2.  **(c)** Aggregation equation §5.3.

**Figure 1: Flows on PDEs with known solution.** We use KDE on the flowed particles for the density plots, and the iterated push-forward density method (Corollary 3.3) to evaluate $\mathcal{F}$ in (a) and (b).

## 5 PDEs with known solutions

We first evaluate our method on gradient flows whose corresponding PDEs have known solutions. We focus on three examples from Carrillo et al. (2021) that combine the three types of functionals introduced in Section 3: porous medium, non-linear Fokker-Planck, and aggregation equations. We tackle 1D versions of these equations in this section, and consider a higher-dimensional ($\mathbb{R}^{20}$) Fokker-Planck equation in Appendix D, where we show that the JKO-ICNN dynamics closely follow Langevin dynamics in recovering the Wasserstein gradient flow corresponding of this PDE (Fig. 3). Throughout this section, we use $\tau = \eta = 10^{-3}$ in the notation of Algorithm 1, and use the ICNN weights from step $t$ to warm-start those of step $t + 1$.

### 5.1 Porous medium equation

The porous medium equation is a classic non-linear diffusion PDE. We consider a diffusion-only system: $\partial_t \rho = \Delta \rho^m, m > 1$, corresponding to a gradient flow of the internal energy $\mathcal{F}(\rho) = \frac{1}{m-1} \int \rho^m(x) \, dx$, which we implement using our JKO-ICNN with objective (12). A known family of exact solutions of this PDE is given by the Barenblatt-Pattle profiles (Zel'dovich & Kompaneetz, 1950; Barenblatt., 1952; Pattle, 1959):

$$\rho(x,t) = t^{-\alpha} \big( C - k\|x\|^2 t^{-2\beta} \big)_+^{\frac{1}{m-1}}, \quad x \in \mathbb{R}^d, \ t > 0,$$

where $C > 0$ is a constant and $\alpha = d/(d(m-1)+2)$, $\beta = \alpha/d$, and $k = \alpha(m-1)/(2md)$.

This exact solution provides a trajectory of densities to compare our JKO-ICNN approach against. Specifically, starting from particles sampled from $\rho(x, 0)$, we can compare the trajectory $\hat{\rho}_t(x)$ estimated with our method to the exact density $\rho(x, t)$. Although this system has no steady-state solution, its asymptotic behavior can be expressed analytically too. For the case $d = 1, m = 2, C = (3/16)^{1/3}$, Figure 1a shows that our method reproduces the dynamics of the exact solution (here the flow density is estimated from particles via KDE and aggregated over 10 repetitions with random initialization) and that the objective value $\mathcal{F}(\hat{\rho})$ has the correct asymptotic behavior.

### 5.2 Nonlinear Fokker-Planck equation

Next, we consider a Fokker-Planck equation with a non-linear diffusion term as before:

$$\partial_t \rho = \nabla \cdot (\rho \nabla V) + \Delta \rho^m, \quad V : \mathbb{R}^d \to \mathbb{R}, \quad m > 1. \tag{13}$$

This PDE corresponds to a gradient flow of the objective $F(\rho) = \frac{1}{m-1} \int \rho^m(x) \, dx + \int V(x) \, d\rho(x)$. For some $V$'s its solutions approach a unique steady state (Carrillo & Toscani, 2000):

$$\rho_\infty(x) = \left(C - \frac{m-1}{m} V(x)\right)_+^{\frac{1}{m-1}}, \tag{14}$$

where the constant $C$ depends on the initial mass of the data. For $d=1$, $m=2$, and $V(x)=x^2$, we solve this PDE using JKO-ICNN with Objectives (10) and (12), using initial data drawn from a Normal distribution with parameters $(\mu, \sigma^2) = (0, 0.2)$. Unlike the previous example, in this case we do not have a full solution $\rho(x, t)$ to compare against, but we can instead evaluate convergence of the flow to $\rho_\infty(x)$. Figure 1b shows that the density $\hat{\rho}_t(x)$ derived from the JKO-ICNN flow converges to the steady state $\rho_\infty(x)$, and so does the value of the objective, i.e., $F(\hat{\rho}_t) \to F(\rho_\infty)$.

### 5.3 Aggregation equation

Next, we consider an aggregation equation: $\partial_t \rho = \nabla \cdot (\rho \nabla W * \rho), W : \mathbb{R}^d \to \mathbb{R}$, which corresponds to a gradient flow on an interaction functional $\mathcal{W}(\rho) = \frac{1}{2} \iint W(x - x') \, d\rho(x) \, d\rho(x')$. We consider the same setting as Carrillo et al. (2021): $d=1$, $\rho_0 \sim \mathcal{N}(0, 1)$, and the kernel $W(x) = \frac{1}{2}|x|^2 - \log(|x|)$, which enforces repulsion at short length scales and attraction at longer scales. This choice of $W$ has the advantage of yielding a unique steady-state equilibrium (Carrillo et al., 2012), given by $\rho_\infty(x) = \frac{1}{\pi}\sqrt{(2 - x^2)_+}$. Our JKO-ICNN encodes $\mathcal{W}$ using Objective (10). As in the previous section, we investigate the convergence of this flow to this steady state distribution. Figure 1c shows that in this case too we observe convergence of densities $\hat{\rho}_t(x) \to \rho_\infty(x)$ and objective values $F(\hat{\rho}_t) \to F(\rho_\infty)$.

## 6 Molecular discovery

To demonstrate the flexibility and efficacy of our approach, we apply it in an important high dimensional setting: controlled generation in molecular discovery. In our experiments, the goal is to increase the *drug-likeness* of a given distribution of molecules while staying close to the original distribution, an important task in drug discovery and drug re-purposing. Formally, given an initial distributions of molecules $\rho_0$, a convex potential energy function $V$ that models the property of interest, and a divergence $D$, we solve:

$$\min_{\rho \in \mathcal{P}(\mathcal{X})} F(\rho) := \lambda_1 \mathbb{E}_\rho V(x) + \lambda_2 D(\rho, \rho_0), \tag{15}$$

We use our JKO-ICNN scheme to optimize this functional on the space of probability measures, given an initial distribution $\rho_0(x) = \frac{1}{N} \sum_{i=1}^{N} \delta_{x_i}(x)$ where $x_i$ is a molecular embedding.

In what follows, we show how we model each component of this functional via: (i) training a molecular embedding using a Variational Auto-encoder (VAE), (ii) training a surrogate potential $V$ to model drug-likeness, (iii) using automatic differentiation via the divergence D.

**Choice of divergence term D** For the divergence D in Equation (15), we use the 2-Wasserstein distance with entropic regularization (Cuturi, 2013).

JKO-ICNN for the functional we optimize in (15) is given for $t \geq 0$ for $\rho_0^\tau = \rho_0$:

$$\theta_{t+1}^\tau \in \underset{\theta : u_\theta \in \text{ICNN}(\mathcal{X})}{\arg\min} \lambda_1 \int V(\nabla_x u_\theta(x)) d\rho_t^\tau + \lambda_2 D((\nabla_x u_\theta(x))_\sharp \rho_t^\tau, \rho_0) + \frac{1}{2\tau} \int_{\mathcal{X}} \|\nabla_x u_\theta(x) - x\|_2^2 \, d\rho_t^\tau$$

$$\rho_{t+1}^\tau = (\nabla_x u_{\theta_{t+1}^\tau})_\# (\rho_t^\tau). \tag{16}$$

We use the Sinkhorn algorithm (Cuturi, 2013) to compute D and backpropagate through this objective as proposed by Genevay et al. (2018), using the geomloss toolbox for efficiency (Feydy et al., 2019) (see Appendix E.2 and E.3 for a discussion of alternative forms of divergence and ablation studies).

**Embedding of molecules using VAEs** We start by training a VAE on string representation of molecules (Polykovskiy et al., 2020; Chenthamarakshan et al., 2020) known as SMILES (Weininger, 1988) to reconstruct these strings with a regularization term that ensures smoothness of the encoder's latent space (Kingma & Welling, 2013; Higgins et al., 2016). We train the VAE on a molecular dataset known as MOSES (Polykovskiy

**Table 2: Comparison between JKO-ICNN and the direct optimization baseline.** For each setup, we report validity, uniqueness, median QED for the final point cloud of embeddings, and Sinkhorn divergence between the initial and final point clouds (Final SD). Each measurement value cell contains mean values $\pm$ one standard deviation for 5 repeated runs with different random initialization seeds, and $\rho_0$ corresponds to the initial point cloud.

| $\lambda_2$ | LR | Validity | Uniqueness | QED Median | Final SD |
|---|---|---|---|---|---|
| $\rho_0$ | | | | | |
| N/A | N/A | $100.000 \pm 0.000$ | $99.980 \pm 0.045$ | $0.630 \pm 0.001$ | N/A |
| *JKO-ICNN* | | | | | |
| $1e^4$ | $1e^{-4}$ | $93.940 \pm 0.336$ | $100.000 \pm 0.000$ | $0.750 \pm 0.001$ | $0.620 \pm 0.010$ |
| *Baseline* - SGD | | | | | |
| 0 | $5e^{-1}$ | $43.440 \pm 1.092$ | $100.000 \pm 0.000$ | $0.772 \pm 0.004$ | $9792.93 \pm 76.913$ |
| 1 | $5e^{-1}$ | $49.440 \pm 1.128$ | $100.000 \pm 0.000$ | $0.768 \pm 0.006$ | $8881.38 \pm 69.736$ |
| $1e^3$ | $5e^{-1}$ | $87.240 \pm 0.777$ | $100.000 \pm 0.000$ | $0.767 \pm 0.002$ | $2515.08 \pm 49.870$ |
| *Baseline* - ADAM | | | | | |
| 0 | $1e^{-1}$ | $92.080 \pm 0.973$ | $100.000 \pm 0.000$ | $0.793 \pm 0.005$ | $18.261 \pm 0.134$ |
| 0 | $1e^{-2}$ | $93.900 \pm 0.781$ | $99.979 \pm 0.048$ | $0.758 \pm 0.006$ | $1.650 \pm 0.006$ |
| 1 | $1e^{-1}$ | $91.200 \pm 0.539$ | $99.978 \pm 0.049$ | $0.792 \pm 0.005$ | $17.170 \pm 0.097$ |
| $1e^3$ | $1e^{-1}$ | $99.980 \pm 0.045$ | $99.980 \pm 0.045$ | $0.630 \pm 0.001$ | $0.077 \pm 0.003$ |
| $1e^4$ | $1e^{-1}$ | $99.900 \pm 0.122$ | $99.980 \pm 0.045$ | $0.630 \pm 0.001$ | $0.240 \pm 0.019$ |

et al., 2020), which is a subset of the ZINC database (Sterling & Irwin, 2015), released under the MIT license. This dataset contains about 1.6M training and 176k test molecules (see Appendix F, for results of this experiment run on a different dataset, QM9 (Ramakrishnan et al., 2014; Ruddigkeit et al., 2012)). Given a molecule, we embed it using the VAE encoder to represent it with a vector $x_i \in \mathbb{R}^{128}$.

**Training a convex surrogate for the desired property (high QED)** The quantitative estimate of drug-likeness (QED) (Bickerton et al., 2012) can be computed with the RDKit library (Landrum, 2013a) but is not differentiable nor convex. Hence, we propose to learn a convex surrogate using Residual ICNNs (Amos et al., 2017; Huang et al., 2021). This ensures that it can be used as convex potential functional $\mathcal{V}$, as described in Section 3. To do so, we process the MOSES dataset via RDKit and obtain a labeled set with QED values. We set a QED threshold of 0.85 and give a lower value label for all QED values above that threshold and a higher value label for all QED values below it so that minimizing the potential functional with this convex surrogate will lead to higher QED values. Given VAE embeddings of the molecules, we train a ICNN classifier on this dataset. See Appendix E.1 for experimental details.

**Optimization with JKO** With the molecule embeddings coming from the VAE serving as the point cloud to be transported and the potential functional defined by the convex QED classifier, we run Algorithm 1 to move an initial point cloud of molecule embeddings $\rho_0$ with low drug-likeness (QED < 0.7) to a region of the latent space that decodes to molecules with distribution $\rho_\tau^T$ with higher drug-likeness. We use the following hyperparameters for JKO-ICNN: $N$, number of original embeddings, was 1,000. The JKO rate $\tau$ was set to $1e^{-4}$ and the outer loop steps $T$ was set to 100. For the inner loop, the number of iterations $n_u$ was set to 500, and the inner loop learning rate $\eta$ was set to $1e^{-3}$. For JKO-ICNN, we used a fully-connected ICNN with two hidden layers, each of dimension 100. Finally, we ran the JKO-ICNN flow without warm starts between steps, as we observed empirically that it resulted in better performance. The full pipeline for this experiment setting is displayed in Figure 4 in Appendix E. All computations were done with 1 CPU and 1 V100 GPU.

**Evaluation** We set $\lambda_1 = 1$ and $\lambda_2 = 10{,}000$ (see Table 4 in Appendix E.3 for details on hyperparameter search). We start JKO with an initial cloud point $\rho_0$ of embeddings that have QED < 0.7 randomly sampled from the MOSES test set. In the second row of Table 2, we see that JKO-ICNN is able to optimize the

functional objective and leads to molecules that satisfy low energy potential, i.e., improved drug-likeness, while staying close to the original point cloud, as measured by Sinkhorn divergence (Final SD).

**Comparison with direct optimization**   We also compare the JKO-ICNN flow to a baseline approach that optimizes the same functional objective via direct gradient descent on the molecule embeddings. Formally, this baseline approach is described by the process: $dX_t^i = -\nabla V(X_t^i)dt - \lambda_2 \nabla D(\frac{1}{N}\sum_{i=1}^N \delta_{X_t^i}, \rho_0), i = 1 \ldots N$, which we discretize using either vanilla gradient descent or ADAM updates. For the baseline, we run a grid search over various hyperparameters and reproduce a selection of configurations in Table 2 where we report validity, uniqueness, median QED (which corresponds to the first term in Equation (15)) of the final point cloud and Sinkhorn divergence between the initial and final point clouds (Final SD, corresponding to the second term in (15)). For the full grid search, see Tables 6 and 7 in Appendix E.6. We note that the only baseline configurations that are able to meaningfully increase median QED are those where $\lambda_2$ is orders of magnitude smaller than in the JKO-ICNN flow. Direct optimization therefore cannot accomplish the joint goals of the objective function. This is consistent with findings by Bunne et al. (2021) with regards to the superiority of JKO over forward methods (direct optimization). This is significant because in many setting it is often crucial to stay close to the original set, e.g., drug re-purposing.

**Benefit of computational amortization**   We show that the maps calculated at each step of the JKO-ICNN flow can be re-used to transport a new set of embeddings with similar gains in QED distribution, without having to retrain the flow (Table 8 in Appendix E.7). We perform this comparison for various sample sizes of initial point clouds and observe linear scaling of the speedup of using the JKO-ICNN map relative to direct optimization. This is a key advantage of JKO-ICNN relative to direct optimization, which needs to be re-optimized for each new set of embeddings.

## 7  Discussion

In this paper, we proposed JKO-ICNN, a scalable method for computing Wasserstein gradient flows. Key to our approach is the parameterization of the space of convex functions with Input Convex Neural Networks. We showed that JKO-ICNN succeeds at optimizing functionals on the space of probability distributions in low-dimensional settings involving known PDES as well as in large-scale and high-dimensional experiments on molecular discovery via controlled generation. Studying the convergence of solutions of JKO-ICNN is an interesting open question that we leave for future work. To mitigate potential risks in biochemical discoveries, generated molecules should be verified in the laboratory, *in vitro* and *in vivo*, before being deployed.

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

## A    Proofs

### A.1    Proof of Lemma 3.1

Starting from the original form of the potential energy functional $\mathcal{V}$ in Equation (3), and using the expression $\rho = (\nabla u)_\sharp \rho_t$ we have:

$$\mathcal{V}(\rho) = \mathcal{V}\big((\nabla u)_\sharp \rho_t\big) = \int V(x)\,\mathrm{d}\,[(\nabla u)_\sharp \rho_t] = \int (V \circ \nabla u)\,\mathrm{d}\rho_t \tag{17}$$

On the other hand, for an interaction functional $\mathcal{W}$ we first note that it can be written as

$$\mathcal{W}(\rho) = \frac{1}{2}\iint W(x-x')\,\mathrm{d}\rho(x')\,\mathrm{d}\rho(x) = \frac{1}{2}\int (W * \rho)(x)\,\mathrm{d}\rho(x). \tag{18}$$

In addition, we will need the fact that

$$W * [(\nabla u)_\sharp \rho] = \int W(x-y)\,\mathrm{d}\,[(\nabla u)_\sharp \rho(y)] = \int W(x - \nabla u(y))\,\mathrm{d}\rho(y). \tag{19}$$

Hence, combining the two equations above we have:

$$\begin{aligned}
\mathcal{W}\big((\nabla u)_\sharp \rho_t\big) &= \frac{1}{2}\int (W * (\nabla u)_\sharp \rho_t)(x)\,\mathrm{d}\,[(\nabla u)_\sharp \rho_t(x)] \\
&= \frac{1}{2}\int \left(\int W(x - \nabla u(y))\,\mathrm{d}\rho_t(y)\right)\mathrm{d}\,[(\nabla u)_\sharp \rho_t(x)] \\
&= \frac{1}{2}\iint W(\nabla u(x) - \nabla u(y))\,\mathrm{d}\rho_t(y)\,\mathrm{d}\rho_t(x),
\end{aligned}$$

as stated. $\square$

### A.2    Proof of Lemma 3.2

Following Santambrogio (2017), we note that whenever $u$ is convex and $\nu$ is absolutely continuous, then $\rho = T_\sharp \nu$ is absolutely continuous too, with a density given by

$$\rho = \frac{\nu}{|\mathbf{J}_T|} \circ T^{-1} \tag{20}$$

where $\mathbf{J}_T$ is the Jacobian matrix of $T$. In our case $\rho = (\nabla u)_\sharp \rho_t$ , so that

$$\rho(y) = \left[\frac{\rho_t}{|\mathbf{H}_u|} \circ (\nabla u)^{-1}\right](y) = \frac{\rho_t\left((\nabla u)^{-1}(y)\right)}{\left|\mathbf{H}_u\left((\nabla u)^{-1}(y)\right)\right|} \tag{21}$$

where $\mathbf{H}$ is the Hessian of $u$. When $u$ is strictly convex it is known that it is invertible and that $(\nabla u)^{-1} = \nabla u^*$, where $u^*$ is the convex conjugate of $u$ (see e.g. Rockafellar (1970)). $\square$

### A.3    Proof of Corollary 3.3

In this proof, we drop the index $\tau$. As before, we use the change of variables $\rho_t = (\nabla u_t)_\sharp \rho_{t-1}$. Thus, by induction,

$$\rho_t = (\nabla u_t \circ \nabla u_t \cdots \circ \nabla u_1)_\sharp \rho_0 \tag{22}$$

Let $T_{1:t} = (\nabla u_t \circ \nabla u_t \cdots \circ \nabla u_1)$, so that $\rho_t = (T_{1:t})_\sharp \rho_0 = (\nabla u_t \circ T_{1:t-1})_\sharp \rho_0$. The Jacobian of this map is given by the chain rule as:

$$\mathbf{J}_{T_{1:t}}(x) = \mathbf{H}_{u_t}(T_{1:t-1}(x))\mathbf{J}_{T_{1:t-1}}(x) \tag{23}$$

Hence by induction we have:

$$\mathbf{J}_{T_{1:t}}(x) = \Pi_{s=1}^{t}\mathbf{H}_{u_s}(T_{1:s-1}(x)) \tag{24}$$

On the other hand, as long as the inverses exist, we have:

$$T_{1:t}^{-1} \quad = \quad (\nabla u_t)^{-1} \circ \cdots \circ (\nabla u_1)^{-1} = \nabla u_t^* \circ \cdots \circ \nabla u_1^*$$

Hence,

$$\rho_t(y) = \left( \frac{\rho_0}{|\mathbf{J}_{T_{1:t}}|} \circ T_{1:t}^{-1} \right)(y) = \left( \frac{\rho_0}{\Pi_{s=1}^{t}|\mathbf{H}_{u_s}(T_{1:s-1})|} \circ T_{1:t}^{-1} \right)(y)$$

$$= \frac{\rho_0\left(T_{1:t}^{-1}(y)\right)}{\Pi_{s=1}^{t}|\mathbf{H}_{u_s}(T_{1:s-1} \circ T_{1:t}^{-1}(y))|},$$

and finally taking the log we obtain:

$$\log\left(\rho_t\right) = \left( \log\left(\rho_0\right) - \sum_{s=1}^{t} \log(|\mathbf{H}_{u_s}(T_{1:s-1})|) \right) \circ (T_{1:t})^{-1}. \qquad \square \tag{25}$$

### A.4  Proof of Lemma 3.4

As before, let $\rho_{t+1} = (\nabla u_{t+1})_\sharp \rho_t$. Thus, by induction,

$$\rho_{t+1} = (\nabla u_{t+1} \circ \nabla u_t \cdots \circ \nabla u_1)_\sharp \rho_0 \tag{26}$$

Let $T_{1:t} = (\nabla u_t \circ \nabla u_t \cdots \circ \nabla u_1)$, so that $\rho_{t+1} = (T_{1:t+1})_\sharp \rho_0 = (\nabla u_{t+1} \circ T_{1:t})_\sharp \rho_0$. The Jacobian of this map is given by the chain rule as:

$$\mathbf{J}_{T_{1:t+1}}(x) = \mathbf{H}_{u_{t+1}}(T_{1:t}(x))\mathbf{J}_{T_{1:t}}(x) \tag{27}$$

On the other hand, using the fact that (whenever the inverses exist) $(f \circ g)^{-1} = g^{-1} \circ f^{-1}$, in our case we have

$$T_{1:t+1}^{-1} = T_{1:t}^{-1} \circ \nabla u_{t+1}^{-1}, \tag{28}$$

Using Equations (20) and (26), we can write the density of $\rho_{t+1}$ as

$$\rho_{t+1}(y) = \left( \frac{\rho_0}{|\mathbf{J}_{T_{1:t+1}}|} \circ T_{1:t+1}^{-1} \right)(y) \tag{29}$$

$$= \frac{\rho_0\left(T_{1:t+1}^{-1}(y)\right)}{|\mathbf{J}_{T_{1:t+1}}\left(T_{1:t+1}^{-1}(y)\right)|} = \frac{\rho_0\left(T_{1:t}^{-1} \circ \nabla u_{t+1}^{-1}(y)\right)}{|\mathbf{H}_{u_{t+1}}(\nabla u_{t+1}^{-1}(y))||\mathbf{J}_{T_{1:t}}(T_{1:t}^{-1} \circ \nabla^{-1}u_{t+1}(y))|} \tag{30}$$

Finally, using the change of variables $y = \nabla u_{t+1} \circ T_{1:t}(x), x = T_{1:t}^{-1} \circ \nabla u_{t+1}^{-1}(y)$, in the integral in the definition of $\mathcal{F}$, we get

$$\mathcal{F}\left(\rho_{t+1}\right) = \int f\left( \frac{\rho_0(x)}{|\mathbf{H}_{u_{t+1}}(T_{1:t}(x))||\mathbf{J}_{T_{1:t}}(x)|} \right) |\mathbf{H}_{u_{t+1}}(T_{1:t}(x))||\mathbf{J}_{T_{1:t}}(x)|\, \mathrm{d}x$$

$$= \mathbb{E}_{x \sim \rho_0}\left[ f\left( \frac{\rho_0(x)}{|\mathbf{H}_{u_{t+1}}(T_{1:t}(x))||\mathbf{J}_{T_{1:t}}(x)|} \right) \frac{|\mathbf{H}_{u_{t+1}}(T_{1:t}(x))||\mathbf{J}_{T_{1:t}}(x)|}{\rho_0(x)} \right]. \qquad \square$$

## B  Practical Considerations

### B.1  Input Convex Neural Networks

Input Convex Neural Networks were introduced by Amos et al. (2017). A $k$-layer fully input convex neural network (FICNN) is one in which each layer has the form:

$$z_{i+1} = g_i\left(W_i^{(z)} z_i + W_i^{(y)} y + b_i\right) \quad i = 0, \ldots, k-1 \tag{31}$$

where $g_i$ are activation functions. Amos et al. (2017) showed that the function $f : x \mapsto z_k$ is convex with respect to $x$ if all the $W_{i:k-1}^{(z)}$ are non-negative, and all the activation functions $g_i$ are convex and non-decreasing. Residual skip connections from the input with linear weights are also allowed and preserve convexity (Amos et al., 2017; Huang et al., 2021).

In our experiments, we parametrize the Brenier potential $u_\theta$ as a FICNN with two hidden layers, with $(100, 20)$ hidden units for the simple PDE experiments in Section 5 and $(100, 100)$ for the molecule generation experiments in Section 6. In order to preserve the convexity of the network, we clip the weights of $W_{i:k-1}^{(z)}$ after every gradient update using $w_{ij} \leftarrow \max\{w_{ij}, 10^{-8}\}$. Alternatively, one can add a small term $+\lambda\|x\|_2^2$ to enforce strong convexity. In all our simple PDE experiments in Section 5, we use the ADAM optimizer with $10^{-3}$ initial learning rate, and a JKO step-size $\tau = 10^{-3}$. Optimization details for the molecular experiments are provided in that section (Section 6).

## B.2 Surrogate loss for entropy

For the choice $f(t) = t \log t$ in the internal energy functional $\mathcal{F}$, we do not use Lemma 3.4 but rather derive the expression from first principles:

$$
\begin{aligned}
\mathcal{F}\big((\nabla_x u_\theta)_\sharp \rho_t\big) &= \int \frac{\rho_t(x)}{|\mathbf{H}_{u_\theta}(x)|} \log \frac{\rho_t(x)}{|\mathbf{H}_{u_\theta}(x)|} |\mathbf{H}_{u_\theta}(x)| \, \mathrm{d}x \\
&= \int \log \frac{\rho_t(x)}{|\mathbf{H}_{u_\theta}(x)|} \rho_t(x) \, \mathrm{d}x \\
&= \int \rho_t(x) \log \rho_t(x) \, \mathrm{d}x - \int \log |\mathbf{H}_{u_\theta}(x)| \rho_t(x) \, \mathrm{d}x = \mathcal{F}(\rho_t) - \mathop{\mathbb{E}}_{x \sim \rho_t} \big[\log |\mathbf{H}_{u_\theta}(x)|\big]
\end{aligned}
$$

As mentioned earlier, this expression has an interesting interpretation as reducing negative entropy (increasing entropy) of $\rho_t$ by an amount given by a log-determinant barrier term on $u$'s Hessian. We see that the only term depending on $\theta$ is $-\mathbb{E}_{x \sim \rho_t}[\log |\mathbf{H}_{u_\theta}(x)|]$. We discuss how to estimate this quantity and backpropagate through $\mathbf{H}_{u_\theta}$ in Appendix B.4.

## B.3 Surrogate losses for internal energies

Let

$$
r_x(u_\theta) = \log(\xi(x)) = \log(\rho_0(x)) - \log |H_{u_\theta}(T_{1:t}(x))| - \log(|J_{1:T}(x)|)
$$

From Lemma 3.4, our point-wise loss is :

$$
L(u_\theta) = \frac{f \circ \exp(r_x(u_\theta))}{\exp(r_x(u_\theta))}
$$

Computing gradient w.r.t $\theta_i$ parameters of $u_\theta$:

$$
\begin{aligned}
\frac{\partial}{\partial \theta_i} L(u) &= \frac{f'(\exp(r_x(u_\theta)))[\exp(r_x(u_\theta))]^2 \frac{\partial}{\partial \theta_i} r_x(u_\theta) - \exp(r_x(u_\theta)) \frac{\partial}{\partial \theta_i} r_x(u_\theta) f \circ \exp(r_x(u_\theta))}{[\exp(r_x(u_\theta)]^2} \\
&= \left( \frac{f'(\exp(r_x(u_\theta))) \exp(r_x(u_\theta)) - f(\exp(r_x(u_\theta)))}{\exp(r_x(u_\theta))} \right) \frac{\partial}{\partial \theta_i} r_x(u_\theta)
\end{aligned}
$$

Also,

$$
\frac{\partial}{\partial \theta_i} r_x(u) = -\frac{\partial}{\partial \theta_i} \log |H_{u_\theta}(T_{1:t}(x))|
$$

Hence the Surrogate loss that has same gradient can be evaluated as follows:

$$
\mathcal{L}(u_\theta) = - \underbrace{\left( \frac{f'(\exp(r_x(u_\theta))) \exp(r_x(u_\theta)) - f(\exp(r_x(u_\theta)))}{\exp(r_x(u_\theta))} \right)}_{\text{no grad}} \log |H_{u_\theta}(T_{1:t}(x))|
$$

For the particular case of porous medium internal energy, let $a = \exp(r)$. For $f(a) = \frac{1}{m-1}a^m = \frac{1}{m-1}\exp(m\log(a)) = \frac{1}{m-1}\exp(mr))$, we have $f'(a) = \frac{m}{m-1}a^{m-1} = \frac{m}{m-1}\exp((m-1)\log(a))$ $f'(a) = \frac{m}{m-1}\exp((m-1)r)$.

$$f'(a) - \frac{f(a)}{a} = \frac{m}{m-1}\exp((m-1)r) - \frac{1}{m-1}\exp((m-1)r) = \exp((m-1)r)$$

Hence we have finally the surrogate loss:

$$\mathcal{L}(u_\theta) = -\underbrace{\exp((m-1)r_x(u_\theta))}_{\text{no grad}}\log|H_{u_\theta}(T_{1:t}(x))|,$$

for which we discuss estimation in Appendix B.4. Table 3 summarizes surrogates losses for common energies in gradient flows.

**Table 3: Surrogate optimization objectives used for computation.** Here $\hat{\mathbb{E}}$ denotes an empirical expectation, $\xi(x)$ is defined in Lemma 3.4, and SG denotes the STOPGRAD operator.

| Functional Type | Exact Form $F(\rho)$ | Surrogate Objective $\hat{F}(u_\theta)$ |
|---|---|---|
| Potential energy | $\int V(x)\,d\rho(x)$ | $\hat{\mathbb{E}}_{x\sim\rho_t}V(\nabla_x u_\theta(x))$ |
| Interaction energy | $\iint W(x-x')\,d\rho(x)\,d\rho(x')$ | $\frac{1}{2}\hat{\mathbb{E}}_{x,y\sim\rho_t}W(\nabla_x u_\theta(x)-\nabla_x u_\theta(y))$ |
| Neg-Entropy | $\int \rho(x)\log\rho(x)\,dx$ | $-\hat{\mathbb{E}}_{x\sim\rho_t}\log|\mathbf{H}_{u_\theta}(x)|$ |
| Nonlinear diffusion | $\int \frac{1}{m-1}\rho(x)^m\,dx$ | $-\hat{\mathbb{E}}_{x\sim\rho_0}e^{(m-1)\text{SG}(\log\xi(x))}\log|\mathbf{H}_{u_\theta}(T_{1:t}(x))|$ |

## B.4 Stochastic log determinant estimators

For numerical reasons, we use different methods to evaluate and compute gradients of Hessian log-determinants.

**Evaluating log-determinants** Following Huang et al. (2021), we use Stochastic Lanczos Quadrature (SLQ) (Ubaru et al., 2017) to estimate the log determinants. We refer to this step as LOGDETESTIMATOR.

**Estimating gradients of log-determinants** The SLQ procedure involves an eigendecomposition, which is unstable to back-propagate through. Thus, to compute gradients, Huang et al. (2021), inspired by Chen et al. (2019), instead use the following expression of the Hessian log-determinant:

$$\frac{\partial}{\partial\theta}\log|\mathbf{H}| = \frac{1}{|\mathbf{H}|}\frac{\partial}{\partial\theta}|\mathbf{H}| = \frac{1}{|\mathbf{H}|}\text{tr}(\text{adj}(\mathbf{H})\frac{\partial H}{\partial\theta}) = \text{tr}(\mathbf{H}^{-1}\frac{\partial H}{\partial\theta}) = \mathbb{E}_v[v^\top\mathbf{H}^{-1}\frac{\partial\mathbf{H}}{\partial\theta}v], \quad (32)$$

Here $v$ is a random Rademacher vector. This last step is the Hutchinson trace estimator (Hutchinson, 1989).

As in Huang et al. (2021), we avoid constructing and inverting the Hessian in this expression by instead solving a problem that requires computing only Hessian-vector products:

$$\underset{z}{\text{argmin}}\,\frac{1}{2}z^\top\mathbf{H}z - v^\top z \quad (33)$$

Since $\mathbf{H}$ is symmetric positive definite, this strictly convex problem has a unique minimizer, $z^*$, that satisfies $z^* = \mathbf{H}^{-1}v$. This problem can be solved using the conjugate gradient method with a fixed number of iterations or a error stopping condition. Thus, computing the last expression in Equation (32) can be done with automatic differentiation by: (i) sampling a Rademacher vector $v$, (ii) running conjugate gradient for $m$ iterations on Problem (33) to obtain $z^m$, (iii) computing $\frac{\partial}{\partial\theta}[(z^m)^\top\mathbf{H}v]$ with automatic differentiation.

**Algorithm 2:** Density estimation for JKO-ICNN

1: **Input:** Query point $x$, sequence of Brenier potentials $\{u_i\}_{i=0}^T$ obtained with JKO-ICNN, initial density evaluation function $\rho_0(\cdot)$.
2: **Initialize** $y_t \leftarrow x$
3: **for** $t = T$ **to** 1 **do**
4:    $y_{t-1} \leftarrow \text{argmax}_y\langle y_t, y\rangle - u_t(y)$
5:    $\{y_{t-1}$ satisfies $(\nabla u_t)(y_{t-1}) = y_t\}$
6: **end for**
7: $x_0 \leftarrow y_0$
8: **for** $t = 0$ **to** $T-1$ **do**
9:    $x_{t+1} \leftarrow \nabla u_t(x_t)$
10: **end for**
11: $\{$Compute $\delta \triangleq \log|\mathbf{J}_{T_{1:t}}(x_0)|\}$
12: $\delta \leftarrow$ LOGDETESTIMATOR$(x_0, x_T)$
13: $p \leftarrow \log\rho_0(y_0) - \delta$
14: **Output:** $p$ satisfying $p = \log\rho_t(x)$

**Complexity of the Hutchinson method** The Hutchinson estimator needs $\mathcal{O}(1/\epsilon^2)$ random projections to give an $1 \pm \epsilon$ guarantee for trace approximation. A newer version, Hutch++ proposed by Meyer et al. (2021), reduces the number of random projections needed to $\mathcal{O}(1/\epsilon)$.

### B.5 Implementation Details

Apart from the stochastic log-determinant estimation (Appendix B.4) needed for computing internal energy functionals, the other main procedure that requires discussion is the density estimation. This is needed, for example, to obtain exact evaluation of the internal energy functionals $\mathcal{F}$ and requires having access to the exact density (or an estimate thereof, e.g., via KDE) from which the initial set of particles were sampled. For this, we rely on Lemma 3.2 and Corollary 3.3, which combined provide a way to estimate the density $\rho_T(x)$ using $\rho_0(x)$, the sequence of Brenier potentials $\{u_i\}_{i=1}^T$, and their combined Hessian log-determinant. This procedure is summarized in Algorithm 2.

## C  Additional qualitative results on 2D datasets

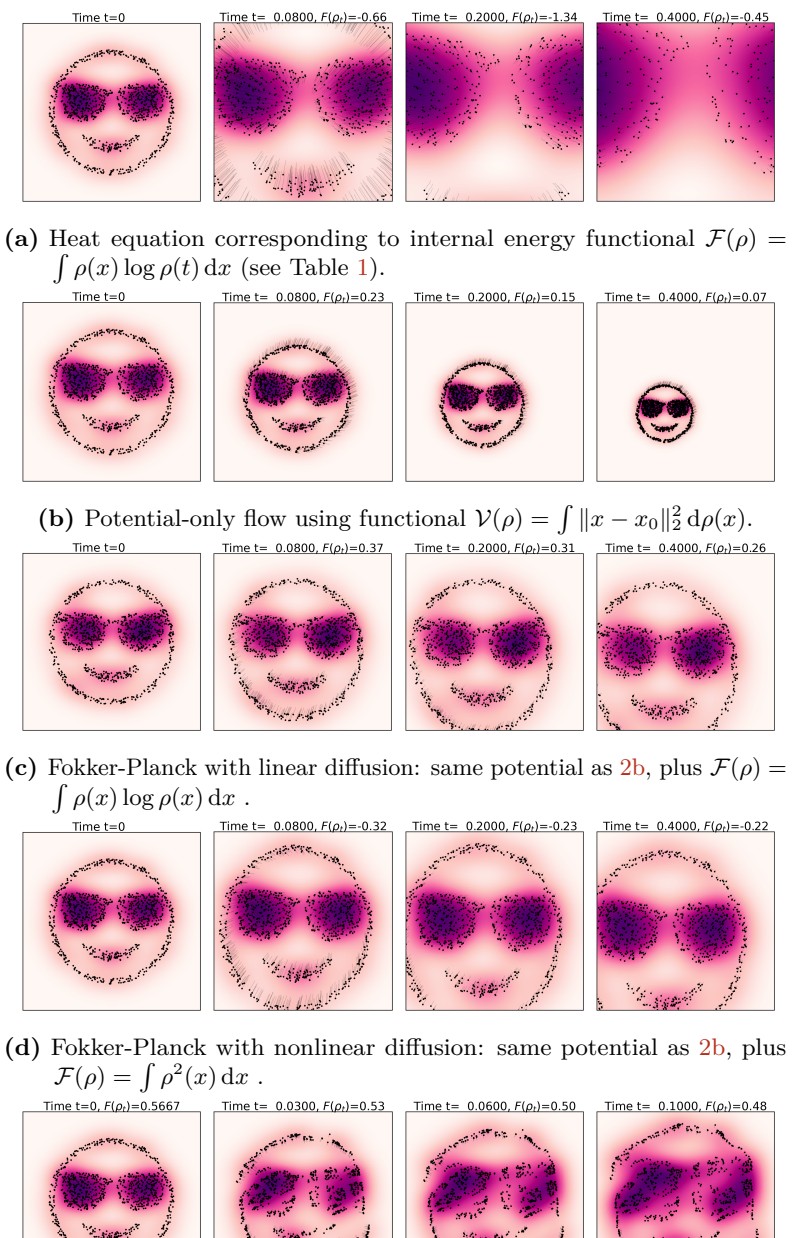

**(a)** Heat equation corresponding to internal energy functional $\mathcal{F}(\rho) = \int \rho(x) \log \rho(t) \, \mathrm{d}x$ (see Table 1).

**(b)** Potential-only flow using functional $\mathcal{V}(\rho) = \int \|x - x_0\|_2^2 \, \mathrm{d}\rho(x)$.

**(c)** Fokker-Planck with linear diffusion: same potential as 2b, plus $\mathcal{F}(\rho) = \int \rho(x) \log \rho(x) \, \mathrm{d}x$ .

**(d)** Fokker-Planck with nonlinear diffusion: same potential as 2b, plus $\mathcal{F}(\rho) = \int \rho^2(x) \, \mathrm{d}x$ .

**(e)** Aggregation equation (same functional as in §5.3).

**Figure 2:** JKO-ICNN flows of a 2D point cloud with density estimated via KDE.

### C.1 Experimental details: PDEs with known solutions

In all the experiments in Section 6, we use the same JKO step-size $\tau = 10^{-3}$ for the outer loop and an ADAM optimizer with initial learning rate $\eta = 10^{-3}$ for the inner loop. We run the inner optimization loop for 400 iterations. In the plots in Figure 1, we show snap-shots at different intervals to facilitate visualization. For these experiments, we parametrize $u_\theta$ as an 2-hidden-layer ICNN with layer width: $(100, 20)$. For the non-linear diffusion term needed for the PDEs in Sections 5.1 and 5.2, we use the surrogate Objective (12). In all cases, we impose strict positivity on the weight matrices $W^{(z)}$ (Equation (31)) with minimum value $\delta = 10^{-18}$ to enforce strong convexity.

## D Comparing high dimensional PDEs to Wasserstein gradient flow

To confirm that JKO-ICNN recovers the Wasserstein gradient flow in high dimension at all time steps, we considered the Fokker-Planck equation and ran the following experiment on the Langevin dynamics in high dimension considering a convex potential: $V(x) = (x - \mu)^\top A(x - \mu)$, where $x, \mu \in \mathbb{R}^d$, $A$ is positive-definite matrix, $F(\rho) = \int V(x)\rho(x) + H(\rho)$, and $H(\rho) = \int \rho \log(\rho)dx$ is the negative entropy.

We chose this functional since Langevin dynamics can be implemented using particles thanks to the Unadjusted Langevin Algorithm (ULA). ULA with learning rate $\eta$ is given by:

$$X_{k+1} = X_k - \eta\nabla V(X_k) + \sqrt{2\eta\beta^{-1}}\xi_k,$$

where $\xi_k \sim \mathcal{N}(0, I_d)$, $\beta$ is a temperature term, and $X_0$ is sampled form $\mathcal{N}(0, \Sigma_0)$. This initial distribution is used for ULA and JKO-ICNN.

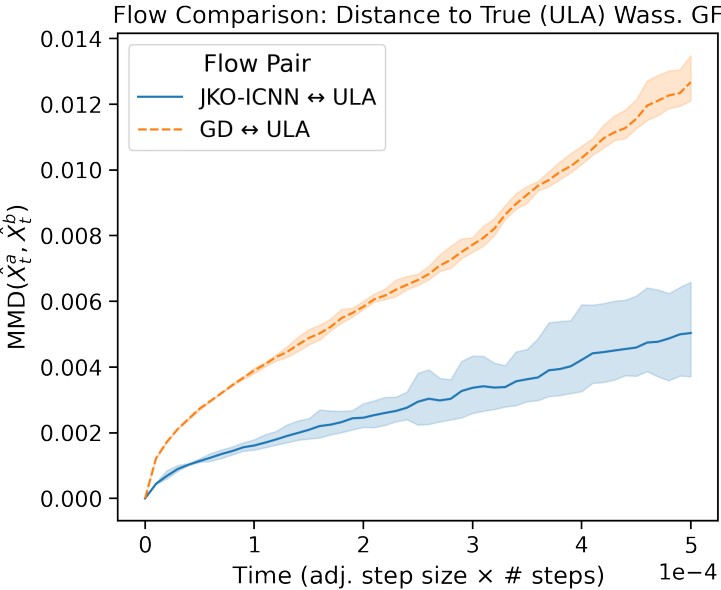

**Figure 3:** Wasserstein Gradient Flow Recovery in high dimension: comparison between the JKO-ICNN flow and direct gradient descent on the objective (GD), evaluated against the 'true' flow obtained by following dynamics using the Unadjusted Langevin Algorithm (ULA) to simulate the Fokker-Planck equation in 20D.

It is known that ULA implements the minimization of $F(\rho)$ using particles, and at the limit of an infinite number of particles and as $\eta \to 0$, the intermediate distribution of $X_k$ of ULA corresponds to the Wasserstein Gradient Flow (WGF) dynamics $\rho_t$. Hence, we compare the distance between the JKO-ICNN intermediate cloud point to ULA's at all times, showing that the JKO-ICNN recovers the Wasserstein gradient flows at all steps (i.e., the MMD of JKO-ICNN's intermediate clouds to ULA's remains small at all time steps, see Figure 3, for $d = 20$).

In order to put these results in context, we also provide the MMD distance of intermediate clouds of ULA versus direct optimization (gradient descent, GD). We see that JKO-ICNN faithfully tracks ULA, relative to GD.

When trying to go beyond $d = 20$, we run into issues with ULA, which is known to become unstable in high dimensions. Although this can be addressed via annealing rates or temperatures, this would make it deviate from the true WGF, defeating the purpose of the comparison, and is out of the scope for this work.

## E  Experimental details: molecular discovery with JKO-ICNN (MOSES dataset)

In what follows, we present the experimental details of the experiments in Section 6, on the MOSES dataset. The MOSES dataset (Polykovskiy et al., 2020) is is a subset of the ZINC database (Sterling & Irwin, 2015). MOSES dataset is available for download at https://github.com/molecularsets/moses, released under the MIT license.

All Molecular discovery JKO-ICNN experiments were run in a compute environment with 1 CPU and 1 V100 GPU submitted as resource-restricted jobs to a cluster. This applies to both convex QED surrogate classifier training and evaluation runs and to the JKO-ICNN flows for each configuration of hyperparameters and random seed initialization. The full pipeline for this experiment is detailed in Figure 4.

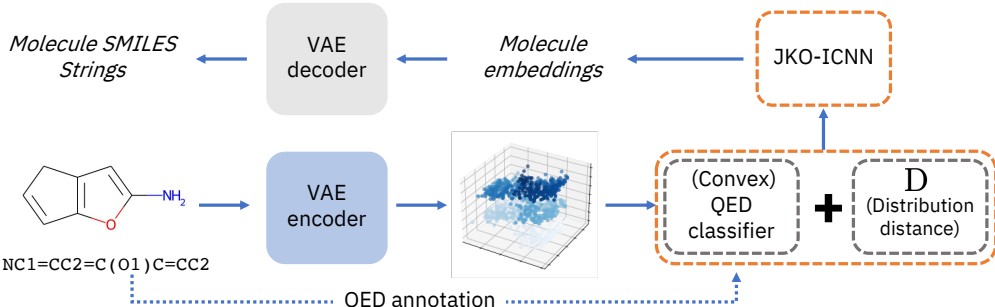

**Figure 4: JKO-ICNN for molecular discovery.** We apply our method to generate molecules whose drug-likeness and closeness to $\rho_0$ are maximized in latent VAE space.

### E.1  Convex QED surrogate classifier

We first describe the hyperparameters and results of the convex surrogate that was trained to predict high ($> 0.85$) and low ($< 0.85$) QED values from molecule embeddings coming from a pre-trained VAE. Molecules with high QED were given lower value labels compared to the low QED molecules so that when this model would be used as potential, minimizing this functional would lead to higher QED values. For this convex surrogate, we trained a Residual ICNN (Amos et al., 2017; Huang et al., 2021) model with four hidden layers, each with dimension 128, which was the dimensionality of the input molecule embeddings as well. We trained the model with binary cross-entropy loss. To maintain convexity in the potential functional however, we used the last layer before the sigmoid activation for $V$. The model was trained with an initial learning rate of 0.01, batch sizes of 1,024, ADAM optimizer, and a learning rate scheduler that decreased learning rate on validation set loss plateau. The model was trained for 100 epochs, and the weights from the final epoch were used to initialize the convex surrogate in the potential functional. For this epoch, the model achieved 85% accuracy on the test set. In Figure 5, we display the test set confusion matrix for this final epoch.

### E.2  Automatic differentiation via D

For the divergence D in Equation (15), we use either the 2-Wasserstein distance with entropic regularization (Cuturi, 2013) or the Maximum Mean Discrepancy (Gretton et al., 2012) with a Gaussian kernel (MMD). When D is the entropy-regularized Wasserstein distance, we use the Sinkhorn algorithm (Cuturi, 2013) to compute D (henceforth denoted as Sinkhorn). We backpropagate through this objective as proposed by Genevay et al. (2018), using the geomloss toolbox for efficiency (Feydy et al., 2019). We also use geomloss for evaluation and backpropagation when D is chosen to be MMD.

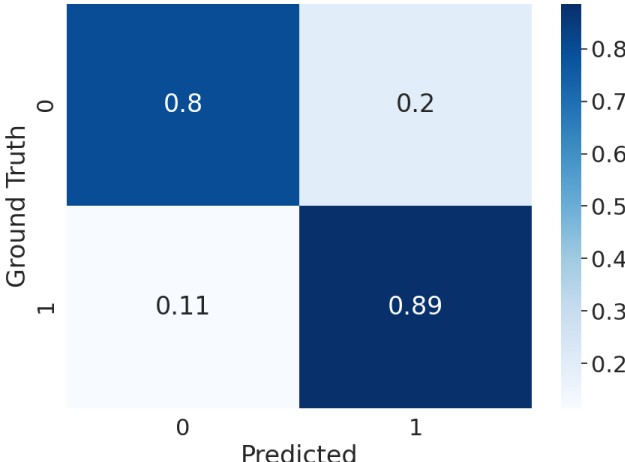

**Figure 5:** Confusion matrix for the convex surrogate trained to predict QED labels from molecule embeddings for MOSES.

### E.3  D ablation study and $\lambda_2$ hyperparameter search

We fixed $\lambda_1 = 1$ for all experiments and used either Sinkhorn or MMD for D. The weight $\lambda_2$ on D was set to either 1,000 or 10,000. We start JKO with an initial cloud point $\rho_0$ of embedding that have QED $< 0.7$ randomly sampled from the MOSES test set. In the Table 4, we report several measurements. First is validity, which is the proportion of the decoded embeddings that have valid SMILES strings according to RDKit. Of the valid strings, we calculate the percent that are unique. Finally, we use RDKit to get the QED annotation of the decoded embeddings and report median values for the point cloud. We re-ran experiments five times with different random seed initializations and report means and standard deviations. In the first row of Table 4, we report the initial values for the point cloud at time $t = 0$. In the last four rows of the table, we report the measurements for different hyperparameter configurations at the end of the JKO scheme for $t = T = 100$. The results are stable across different runs as can be seen from the reported standard deviations from the different random initializations. We notice that Sinkhorn divergence with $\lambda_2 =$10,000 prevents mode collapse and preserves uniqueness of the transported embedding via JKO-ICNN. While MMD yields higher drug-likeness, it leads to a deterioration in uniqueness. Using Sinkhorn allows for a matching between the transformed point cloud and the original one, which preserves better uniqueness than MMD, which merely matches mean embeddings of the distributions. The JKO-ICNN experiment reported in Table 2 in Section 6 and in Table 8 uses Sinkhorn as the divergence term.

### E.4  $\tau$ hyperparameter search

Our search of optimal $\tau$ was done under the following setup: learning rate for training the ICNN was fixed at $\eta = 0.001$. $\lambda_2$ set to 10,000, the JKO outer loop was run for 100 steps, and inner loop optimization was set to 500 steps. The results are presented in Table 5. We notice that unlike direct optimization (see Tables 6 and 7), JKO-ICNN is robust across learning rates $\tau$.

### E.5  JKO-ICNN QED histograms

In Figure 6, we present several time steps of the histograms of the QED values for the decoded SMILES strings corresponding to the point clouds for the experiment that used Sinkhorn as the distribution distance with weight 10,000.

### E.6  Direct optimization baseline hyperparameter grid search

The full grid search over hyperparameters for the direct optimization baseline discussed in Section 6 is available in Tables 6 and 7. In order to ensure a 'fair shot' at competing with our JKO-ICNN approach, we performed a grid search over the following hyperparameters:

**Table 4: Molecular discovery with JKO-ICNN experiment results** Measures of validity, uniqueness, and median, average, and standard deviation QED are reported in each row for the corresponding point cloud of embeddings. For each point cloud, we decode the embeddings to get SMILES strings and use RDKit to determine whether the corresponding string is valid and get the associated QED value. In the first row, we display the point cloud at time step zero of JKO-ICNN. In the subsequent rows, we display the values for each measurement at the final time step $T$ of JKO-ICNN for different hyperparameter configurations of distribution distance D (either Sinkhorn or MMD) and weight on this distance $\lambda_2$ (either 1,000 or 10,000). Each measurement value cell contains mean values $\pm$ one standard deviation for five repeated runs of the experiment with different random initialization seeds. We find that the setup that uses Sinkhorn with weight 10,000 yields the best results in terms of moving the point cloud towards regions with higher QED without sacrificing validity and uniqueness of the decoded SMILES strings. This table contains results for the MOSES dataset.

| Measure | D | $\lambda_2$ | Validity | Uniqueness | QED Median | QED Avg. | QED Std. |
|---------|---|---|---|---|---|---|---|
| $\rho_0$ | N/A | N/A | $100.000 \pm 0.000$ | $99.980 \pm 0.045$ | $0.630 \pm 0.001$ | $0.621 \pm 0.000$ | $0.063 \pm 0.002$ |
| $\rho_T^\tau$ | Sinkhorn | 1,000 | $92.460 \pm 2.096$ | $69.919 \pm 4.906$ | $0.746 \pm 0.016$ | $0.735 \pm 0.009$ | $0.110 \pm 0.003$ |
| $\rho_T^\tau$ | Sinkhorn | 10,000 | $93.020 \pm 1.001$ | $99.245 \pm 0.439$ | $0.769 \pm 0.002$ | $0.754 \pm 0.003$ | $0.112 \pm 0.002$ |
| $\rho_T^\tau$ | MMD | 1,000 | $94.560 \pm 1.372$ | $51.668 \pm 2.205$ | $0.780 \pm 0.009$ | $0.767 \pm 0.013$ | $0.107 \pm 0.012$ |
| $\rho_T^\tau$ | MMD | 10,000 | $92.020 \pm 3.535$ | $53.774 \pm 3.013$ | $0.776 \pm 0.014$ | $0.767 \pm 0.009$ | $0.102 \pm 0.011$ |

**Table 5: Molecular discovery with JKO-ICNN experiment $\tau$ hyperparameter search.** Measures of validity, uniqueness, and median QED of final point cloud of embeddings are presented for each $\tau$. Each measurement value cell contains mean values $\pm$ one standard deviation for five repeated runs of the experiment with different random initialization seeds. For this search, we fix the other hyperparameters: $\eta = 0.001$, $\lambda_2 = 10,000$, the JKO outer loop was run for 100 steps, and inner loop optimization was set to 500 steps. This table contains results for the MOSES dataset.

| $\tau$ | Validity | Uniqueness | QED Median |
|--------|----------|------------|------------|
| 0.01 | $94.600 \pm 0.620$ | $99.979 \pm 0.047$ | $0.708 \pm 0.007$ |
| 0.001 | $94.620 \pm 0.907$ | $99.979 \pm 0.047$ | $0.716 \pm 0.005$ |
| 0.0001 | $93.320 \pm 0.687$ | $99.957 \pm 0.059$ | $0.751 \pm 0.007$ |

- Optimizer: {SGD, ADAM}

- Learning rate (LR): $\{5e^{-1}, 1e^{-1}, 1e^{-2}, 1e^{-3}, 1e^{-4}\}$

- $\lambda_2$: $\{0, 1, 10, 100, 1,000, 10,000\}$

and report validity, uniqueness, median QED of the final point cloud and Sinkhorn divergence between the initial and final point clouds (Final SD).

### E.7 Computation amortization results

In Table 8, we compare the results and computation time for the the best direct optimization hyperparameter configuration found during the grid search (Optimizer: ADAM, LR: 0.1, $\lambda_2$: 1) vs. re-using the maps found during the JKO-ICNN flow applied to a new set of embeddings. We observe linear scaling speed-up when reusing the JKO-ICNN maps vs. re-optimizing the baseline on a new set of points.

## F Molecule JKO-ICNN experimental setup and results for QM9 dataset

In this section, we repeat the results and analysis presentation of Appendix E but for the experiments that used the QM9 dataset (Ramakrishnan et al., 2014; Ruddigkeit et al., 2012). The QM9 dataset contains on average smaller molecules than MOSES. The MOSES dataset is larger in size than QM9 (Training: 1.6M molecules in MOSES vs. 121k in QM9; Test: 176k in MOSES molecules vs. 13k in QM9). For these experiments a separate VAE model and convex surrogate classifier were trained using the QM9 dataset.

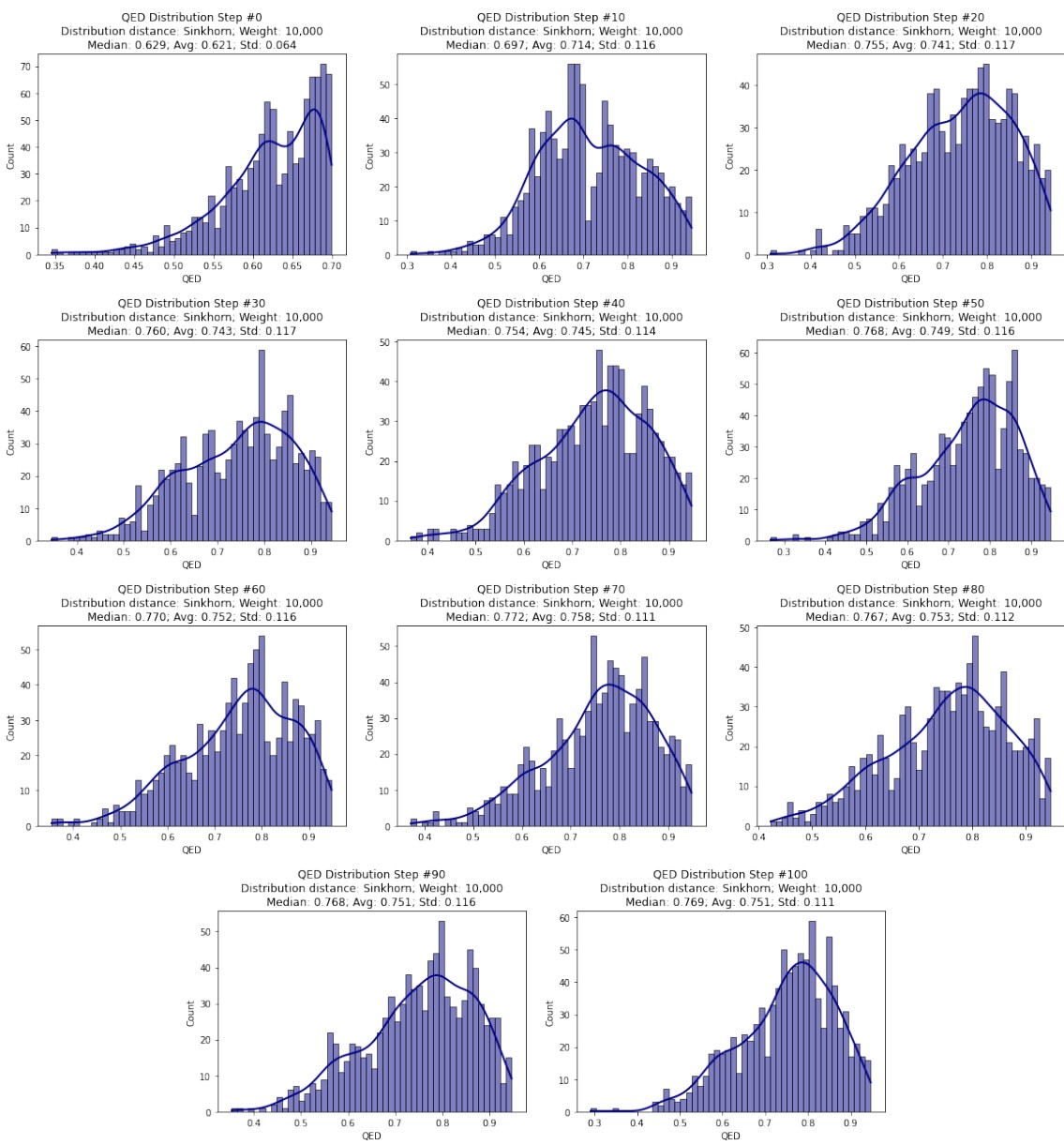

**Figure 6:** Histograms of QED values for the decoded SMILES strings corresponding to the point clouds at time step $t = 0, 10, 20...100$ for the JKO-ICNN experiment that uses Sinkhorn as the distribution distance with weight 10,000. We observe a clear shift to the right in the distribution, which corresponds to increased drug-likeness of the decoded molecule strings. This figure displays results for the MOSES dataset experiment.

## F.1 Convex QED surrogate classifier (QM9)

For QM9, the convex surrogate was trained to predict high ($> 0.5$) and low ($< 0.5$) QED values from molecule embeddings coming from a pre-trained VAE. The threshold for QM9 molecules was set to a lower value compared to that for the MOSES dataset because the underlying QED distribution of train and test data for QM9 molecules has significantly lower values compared to MOSES. As above, molecules with high QED were given lower value labels compared to the low QED molecules so that when this model would be used as potential, minimizing this functional would lead to higher QED values. Similar to the MOSES dataset pipeline, for this convex surrogate, we trained a Residual ICNN (Amos et al., 2017; Huang et al., 2021) model with four hidden layers, each with dimension 128, which was the dimensionality of the input molecule

**Table 6:** Molecular discovery with direct optimization of functional objective using the **SGD** optimizer: Measures of validity, uniqueness, and median QED are reported in each row for the corresponding final point cloud of embeddings in each configuration. We also report the final Sinkhorn divergence between the initial cloud point and that from the final time step (Final SD). Each measurement value cell contains mean values ± one standard deviation for five repeated runs of the experiment with different random initialization seeds. This table contains results for the MOSES dataset. † For the SGD optimizer, the only configurations that meaningfully increase the QED distribution of the given point cloud are those with relatively large learning rate, 0.5. However, this leads the point cloud to regions that decode to invalid molecule strings and Final SD being several orders of magnitude higher compared to the JKO-ICNN approach.

| *Grid search for* SGD *optimizer* | | | | | |
|---|---|---|---|---|---|
| $\lambda_2$ | LR | Validity | Uniqueness | QED Median | Final SD |
| 0 | $0.5^\dagger$ | $43.440 \pm 1.092$ | $100.000 \pm 0.000$ | $0.772 \pm 0.004$ | $9792.929 \pm 76.913$ |
| 0 | 0.1 | $99.960 \pm 0.055$ | $99.980 \pm 0.045$ | $0.630 \pm 0.002$ | $4.584 \pm 1.182$ |
| 0 | 0.01 | $99.960 \pm 0.055$ | $99.980 \pm 0.045$ | $0.630 \pm 0.001$ | $0.022 \pm 0.000$ |
| 0 | 0.001 | $99.980 \pm 0.045$ | $99.980 \pm 0.045$ | $0.630 \pm 0.001$ | $0.017 \pm 0.000$ |
| 0 | 0.0001 | $100.000 \pm 0.000$ | $99.980 \pm 0.045$ | $0.630 \pm 0.001$ | $0.017 \pm 0.000$ |
| 1 | $0.5^\dagger$ | $49.440 \pm 1.128$ | $100.000 \pm 0.000$ | $0.768 \pm 0.006$ | $8881.378 \pm 69.736$ |
| 1 | 0.1 | $99.960 \pm 0.055$ | $99.980 \pm 0.045$ | $0.630 \pm 0.002$ | $4.496 \pm 1.156$ |
| 1 | 0.01 | $99.960 \pm 0.055$ | $99.980 \pm 0.045$ | $0.630 \pm 0.001$ | $0.022 \pm 0.000$ |
| 1 | 0.001 | $99.980 \pm 0.045$ | $99.980 \pm 0.045$ | $0.630 \pm 0.001$ | $0.017 \pm 0.000$ |
| 1 | 0.0001 | $100.000 \pm 0.000$ | $99.980 \pm 0.045$ | $0.630 \pm 0.001$ | $0.017 \pm 0.000$ |
| 10 | $0.5^\dagger$ | $83.660 \pm 0.918$ | $100.000 \pm 0.000$ | $0.771 \pm 0.007$ | $3696.657 \pm 28.212$ |
| 10 | 0.1 | $99.960 \pm 0.055$ | $99.980 \pm 0.045$ | $0.630 \pm 0.002$ | $3.832 \pm 0.917$ |
| 10 | 0.01 | $99.960 \pm 0.055$ | $99.980 \pm 0.045$ | $0.630 \pm 0.001$ | $0.021 \pm 0.000$ |
| 10 | 0.001 | $99.980 \pm 0.045$ | $99.980 \pm 0.045$ | $0.630 \pm 0.001$ | $0.017 \pm 0.000$ |
| 10 | 0.0001 | $100.000 \pm 0.000$ | $99.980 \pm 0.045$ | $0.630 \pm 0.001$ | $0.017 \pm 0.000$ |
| 100 | $0.5^\dagger$ | $96.920 \pm 0.576$ | $98.885 \pm 0.308$ | $0.635 \pm 0.002$ | $547.310 \pm 36.557$ |
| 100 | 0.1 | $99.960 \pm 0.055$ | $99.980 \pm 0.045$ | $0.630 \pm 0.002$ | $0.822 \pm 0.125$ |
| 100 | 0.01 | $99.960 \pm 0.055$ | $99.980 \pm 0.045$ | $0.630 \pm 0.001$ | $0.021 \pm 0.000$ |
| 100 | 0.001 | $99.980 \pm 0.045$ | $99.980 \pm 0.045$ | $0.630 \pm 0.001$ | $0.017 \pm 0.000$ |
| 100 | 0.0001 | $100.000 \pm 0.000$ | $99.980 \pm 0.045$ | $0.630 \pm 0.001$ | $0.017 \pm 0.000$ |
| 1,000 | $0.5^\dagger$ | $87.240 \pm 0.777$ | $100.000 \pm 0.000$ | $0.767 \pm 0.002$ | $2515.075 \pm 49.870$ |
| 1,000 | 0.1 | $99.960 \pm 0.055$ | $99.980 \pm 0.045$ | $0.630 \pm 0.002$ | $1.833 \pm 0.611$ |
| 1,000 | 0.01 | $99.960 \pm 0.055$ | $99.980 \pm 0.045$ | $0.630 \pm 0.001$ | $0.019 \pm 0.000$ |
| 1,000 | 0.001 | $99.980 \pm 0.045$ | $99.980 \pm 0.045$ | $0.630 \pm 0.001$ | $0.017 \pm 0.000$ |
| 1,000 | 0.0001 | $100.000 \pm 0.000$ | $99.980 \pm 0.045$ | $0.630 \pm 0.001$ | $0.017 \pm 0.000$ |
| 10,000 | 0.5 | $nan \pm nan$ | $nan \pm nan$ | $nan \pm nan$ | $nan \pm nan$ |
| 10,000 | 0.1 | $98.500 \pm 0.235$ | $99.980 \pm 0.045$ | $0.641 \pm 0.003$ | $296.809 \pm 9.344$ |
| 10,000 | 0.01 | $99.980 \pm 0.045$ | $99.980 \pm 0.045$ | $0.630 \pm 0.001$ | $0.017 \pm 0.000$ |
| 10,000 | 0.001 | $99.980 \pm 0.045$ | $99.980 \pm 0.045$ | $0.630 \pm 0.001$ | $0.017 \pm 0.000$ |
| 10,000 | 0.0001 | $100.000 \pm 0.000$ | $99.980 \pm 0.045$ | $0.630 \pm 0.001$ | $0.017 \pm 0.000$ |

embeddings as well. We trained the model with binary cross-entropy loss. To maintain convexity in the potential functional, we used the last layer before the sigmoid activation for $V$. The model was trained with an initial learning rate of 0.01, batch sizes of 1,024, ADAM optimizer, and a learning rate scheduler that decreased learning rate on validation set loss plateau. The model was trained for 100 epochs, and the weights from the final epoch were used to initialize the convex surrogate in the potential functional. For this epoch, the model achieved 88.6% accuracy on the test set. In Figure 7, we display the test set confusion matrix for this final epoch.

### F.2   JKO-ICNN QED histograms (QM9)

In Table 9, we present the same results as in Appendix E.5. For the JKO-ICNN flow on the QM9 dataset, we started with an initial distribution of embeddings $\rho_0$ that had corresponding QED value of $< 0.35$. This initial point cloud was taken from the QM9 train set since the test set is quite small and does not contain enough data points below the starting QED threshold.

As seen with the experiment on MOSES, all four combinations are able to increase QED values. However, the setups that use MMD or $\lambda_2 = 1,000$ lead to mode collapse, see the discussion in Appendix E.3.

In Figure 8, we present several time steps of the histograms of the QED values for the decoded SMILES strings corresponding to the point clouds for the experiment that used Sinkhorn as the distribution distance with weight 10,000.

**Table 7:** Molecular discovery with direct optimization of functional objective using the **ADAM** optimizer: Measures of validity, uniqueness, and median QED are reported in each row for the corresponding final point cloud of embeddings in each configuration. We also report the final Sinkhorn divergence between the initial cloud point and that from the final time step (Final SD). Each measurement value cell contains mean values $\pm$ one standard deviation for five repeated runs of the experiment with different random initialization seeds. This table contains results for the MOSES dataset. † For the ADAM optimizer, although there are some hyperparameter configurations that are able to meaningfully increase QED of the point cloud without sacrificing the validity of the decoded molecule strings, these configurations require $\lambda_2$ to either be 0 or several orders of magnitude lower compared to in the JKO-ICNN approach. This direct optimization approach is therefore unable to successfully meet both goals of the objective, higher QED with small distance from the original point cloud, as seen by the larger Final SD values.

| *Grid search for* ADAM *optimizer* | | | | | |
|---|---|---|---|---|---|
| $\lambda_2$ | LR | Validity | Uniqueness | QED Median | Final SD |
| 0 | 0.5 | $7.420 \pm 0.729$ | $98.479 \pm 1.595$ | $0.654 \pm 0.016$ | $444.779 \pm 1.168$ |
| 0 | $0.1^\dagger$ | $92.080 \pm 0.973$ | $100.000 \pm 0.000$ | $0.793 \pm 0.005$ | $18.261 \pm 0.134$ |
| 0 | $0.01^\dagger$ | $93.900 \pm 0.781$ | $99.979 \pm 0.048$ | $0.758 \pm 0.006$ | $1.650 \pm 0.006$ |
| 0 | 0.001 | $99.320 \pm 0.164$ | $99.980 \pm 0.045$ | $0.632 \pm 0.001$ | $0.073 \pm 0.000$ |
| 0 | 0.0001 | $99.900 \pm 0.100$ | $99.980 \pm 0.045$ | $0.630 \pm 0.002$ | $0.018 \pm 0.000$ |
| 1 | 0.5 | $46.240 \pm 1.553$ | $100.000 \pm 0.000$ | $0.740 \pm 0.005$ | $394.993 \pm 1.093$ |
| 1 | $0.1^\dagger$ | $91.200 \pm 0.539$ | $99.978 \pm 0.049$ | $0.792 \pm 0.005$ | $17.170 \pm 0.097$ |
| 1 | $0.01^\dagger$ | $95.100 \pm 0.505$ | $99.979 \pm 0.047$ | $0.702 \pm 0.004$ | $1.551 \pm 0.009$ |
| 1 | 0.001 | $99.340 \pm 0.167$ | $99.980 \pm 0.045$ | $0.632 \pm 0.001$ | $0.072 \pm 0.000$ |
| 1 | 0.0001 | $99.900 \pm 0.100$ | $99.980 \pm 0.045$ | $0.630 \pm 0.002$ | $0.018 \pm 0.000$ |
| 10 | 0.5 | $96.940 \pm 0.182$ | $92.635 \pm 0.496$ | $0.656 \pm 0.003$ | $241.505 \pm 1.616$ |
| 10 | 0.1 | $95.380 \pm 0.618$ | $99.937 \pm 0.057$ | $0.701 \pm 0.005$ | $13.360 \pm 0.030$ |
| 10 | 0.01 | $98.900 \pm 0.464$ | $99.980 \pm 0.045$ | $0.637 \pm 0.002$ | $1.253 \pm 0.005$ |
| 10 | 0.001 | $99.680 \pm 0.164$ | $99.980 \pm 0.045$ | $0.631 \pm 0.001$ | $0.068 \pm 0.000$ |
| 10 | 0.0001 | $99.900 \pm 0.100$ | $99.980 \pm 0.045$ | $0.630 \pm 0.001$ | $0.018 \pm 0.000$ |
| 100 | 0.5 | $99.680 \pm 0.084$ | $90.610 \pm 0.783$ | $0.631 \pm 0.002$ | $18.000 \pm 0.605$ |
| 100 | 0.1 | $99.840 \pm 0.182$ | $99.960 \pm 0.055$ | $0.630 \pm 0.001$ | $4.528 \pm 0.068$ |
| 100 | 0.01 | $99.920 \pm 0.084$ | $99.980 \pm 0.045$ | $0.630 \pm 0.001$ | $0.246 \pm 0.000$ |
| 100 | 0.001 | $99.980 \pm 0.045$ | $99.980 \pm 0.045$ | $0.630 \pm 0.001$ | $0.054 \pm 0.000$ |
| 100 | 0.0001 | $99.980 \pm 0.045$ | $99.980 \pm 0.045$ | $0.630 \pm 0.002$ | $0.018 \pm 0.000$ |
| 1,000 | 0.5 | $96.140 \pm 0.279$ | $99.105 \pm 0.262$ | $0.662 \pm 0.006$ | $12.146 \pm 0.547$ |
| 1,000 | 0.1 | $99.980 \pm 0.045$ | $99.980 \pm 0.045$ | $0.630 \pm 0.001$ | $0.077 \pm 0.003$ |
| 1,000 | 0.01 | $99.940 \pm 0.055$ | $99.980 \pm 0.045$ | $0.630 \pm 0.001$ | $0.023 \pm 0.000$ |
| 1,000 | 0.001 | $99.980 \pm 0.045$ | $99.980 \pm 0.045$ | $0.630 \pm 0.001$ | $0.021 \pm 0.000$ |
| 1,000 | 0.0001 | $99.980 \pm 0.045$ | $99.980 \pm 0.045$ | $0.630 \pm 0.001$ | $0.018 \pm 0.000$ |
| 10,000 | 0.5 | $99.020 \pm 0.295$ | $97.697 \pm 0.221$ | $0.635 \pm 0.002$ | $2.646 \pm 0.062$ |
| 10,000 | 0.1 | $99.900 \pm 0.122$ | $99.980 \pm 0.045$ | $0.630 \pm 0.001$ | $0.240 \pm 0.019$ |
| 10,000 | 0.01 | $99.980 \pm 0.045$ | $99.980 \pm 0.045$ | $0.630 \pm 0.001$ | $0.018 \pm 0.000$ |
| 10,000 | 0.001 | $99.980 \pm 0.045$ | $99.980 \pm 0.045$ | $0.630 \pm 0.001$ | $0.017 \pm 0.000$ |
| 10,000 | 0.0001 | $99.980 \pm 0.045$ | $99.980 \pm 0.045$ | $0.630 \pm 0.001$ | $0.017 \pm 0.000$ |

## G Assets

**Software** Our implementation of JKO-ICNN relies on various open-source libraries, including pytorch (Paszke et al., 2019) (license: BSD), pytorch-lightning (Falcon et al., 2019) (Apache 2.0), POT (Flamary et al., 2021) (MIT), geomloss (Feydy et al., 2019) (MIT), rdkit (Landrum, 2013a;b) (BSD 3-Clause).

**Data** All data used in Section 5 is synthetic. The MOSES dataset is released under the MIT license. The QM9 dataset does not explicitly provide a license in their website nor data files.

**Table 8: Computational amortization gains.** Comparison between re-use of maps calculated from JKO-ICNN flow and the direct optimization baseline applied to various sizes of embedding point clouds. For each setup, we report median QED for the final point cloud of embeddings from the respective flows and the Sinkhorn divergence between the initial and final point clouds (Final SD). We observe a linear trend in speed-up when re-using JKO-ICNN maps. This table contains results for the MOSES dataset experiment.

| Size | QED Median | Final SD | Time (s) | Speedup |
|---|---|---|---|---|
| *Baseline* | | | | |
| 1,000 | 0.789 | 17.134 | 1.416 | — |
| 2,000 | 0.795 | 17.333 | 1.550 | — |
| 3,000 | 0.791 | 17.255 | 2.300 | — |
| 4,000 | 0.790 | 17.238 | 3.433 | — |
| 5,000 | 0.792 | 17.212 | 4.696 | — |
| *JKO-ICNN maps* | | | | |
| 1,000 | 0.778 | 1.739 | 0.086 | 1.330 |
| 2,000 | 0.773 | 1.440 | 0.086 | 1.464 |
| 3,000 | 0.771 | 1.363 | 0.086 | 2.213 |
| 4,000 | 0.767 | 1.373 | 0.086 | 3.347 |
| 5,000 | 0.764 | 1.384 | 0.088 | 4.608 |

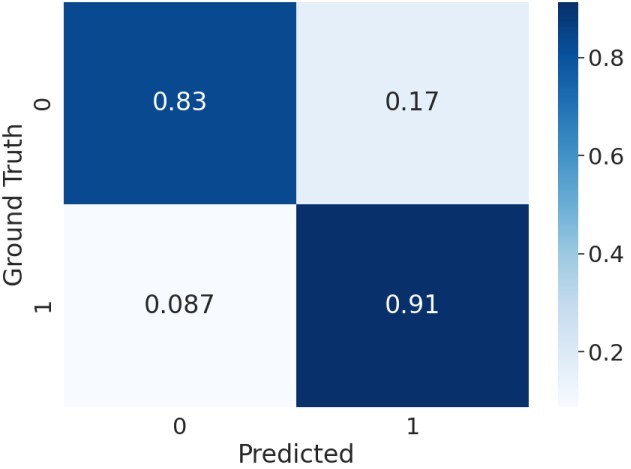

**Figure 7:** Confusion matrix for the convex surrogate trained to predict QED labels from molecule embeddings for QM9.

**Table 9: Molecular discovery with JKO-ICNN experiment results (QM9)** Measures of validity, uniqueness, and median, average, and standard deviation QED are reported in each row for the corresponding point cloud of embeddings. For each point cloud, we decode the embeddings to get SMILES strings and use RDKit to determine whether the corresponding string is valid and get the associated QED value. In the first row, we display the point cloud at time step zero of JKO-ICNN. In the subsequent rows, we display the values for each measurement at the final time step $T$ of JKO-ICNN for different hyperparameter configurations of distribution distance D (either Sinkhorn or MMD) and weight on this distance $\lambda_2$ (either 1,000 or 10,000). Each measurement value cell contains mean values $\pm$ one standard deviation for five repeated runs of the experiment with different random initialization seeds. This table contains results for the QM9 dataset experiment.

| Measure | D | $\lambda_2$ | Validity | Uniqueness | QED Median | QED Avg. | QED Std. |
|---------|---|-------------|----------|------------|------------|----------|----------|
| $\rho_0$ | N/A | N/A | $100.000 \pm 0.000$ | $99.840 \pm 0.134$ | $0.315 \pm 0.001$ | $0.303 \pm 0.001$ | $0.041 \pm 0.001$ |
| $\rho_T^\tau$ | Sinkhorn | 1,000 | $90.840 \pm 1.457$ | $33.367 \pm 2.491$ | $0.381 \pm 0.024$ | $0.373 \pm 0.016$ | $0.096 \pm 0.005$ |
| $\rho_T^\tau$ | Sinkhorn | 10,000 | $92.700 \pm 0.828$ | $81.925 \pm 1.982$ | $0.419 \pm 0.005$ | $0.404 \pm 0.005$ | $0.096 \pm 0.004$ |
| $\rho_T^\tau$ | MMD | 1,000 | $91.680 \pm 4.463$ | $22.424 \pm 1.185$ | $0.452 \pm 0.024$ | $0.434 \pm 0.019$ | $0.094 \pm 0.005$ |
| $\rho_T^\tau$ | MMD | 10,000. | $88.800 \pm 4.661$ | $28.664 \pm 1.500$ | $0.448 \pm 0.013$ | $0.432 \pm 0.010$ | $0.093 \pm 0.005$ |

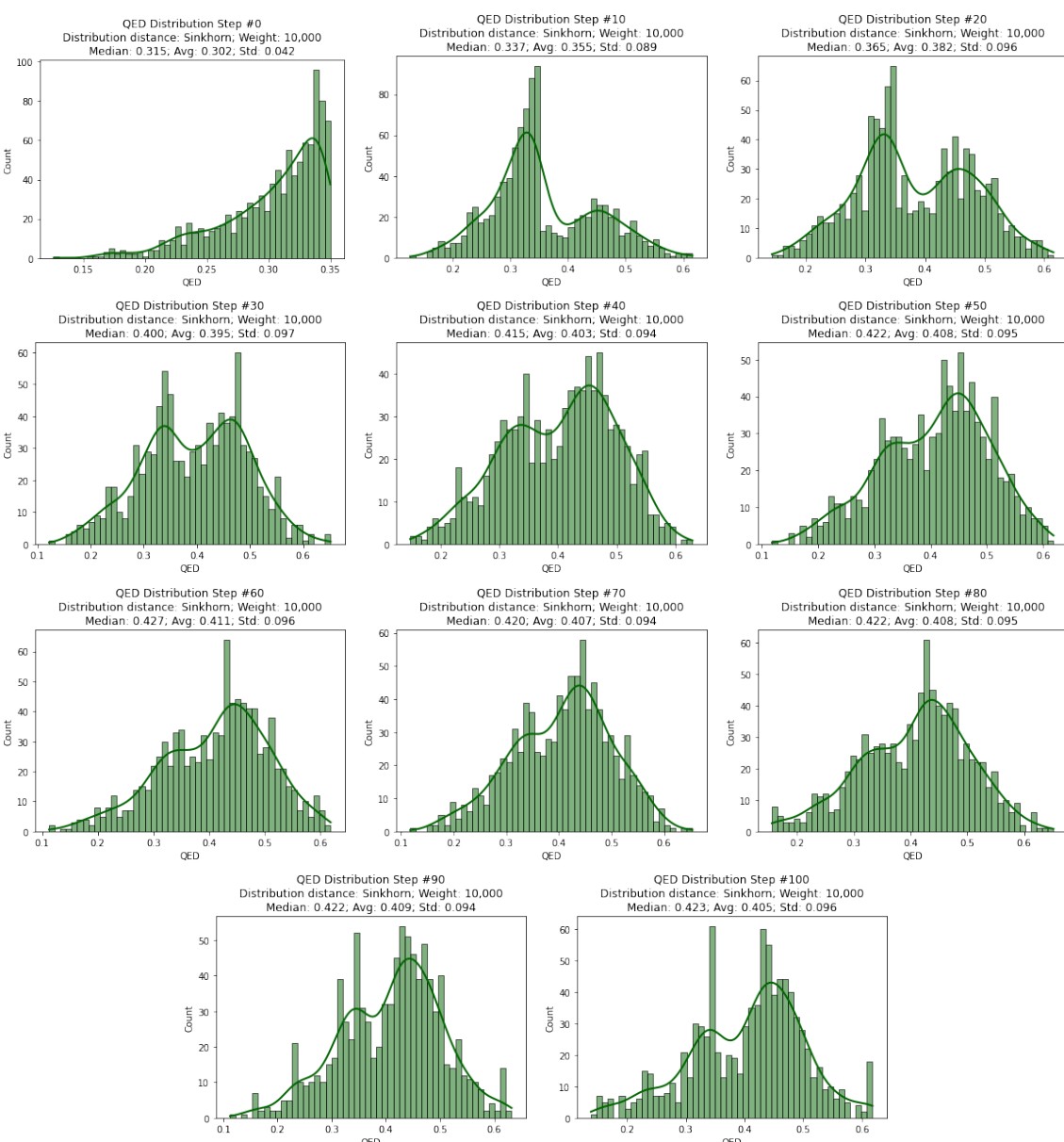

**Figure 8:** Histograms of QED values for the decoded SMILES strings corresponding to the point clouds at time step $t = 0, 10, 20...100$ for the JKO-ICNN experiment that uses Sinkhorn as the distribution distance with weight 10,000. We observe a clear shift to the right in the distribution, which corresponds to increased drug-likeness of the decoded molecule strings. This figure displays results for the QM9 dataset experiment.

