# OpenReview forum: "Optimizing Functionals on the Space of Probabilities with Input Convex Neural Networks"
_TMLR — Accepted by TMLR_

### Review · Reviewer_x751 · 2022-05-08

**Summary Of Contributions:**

This paper deals with the optimization of functionals on the space of probability measures. Following a gradient flow approach, the authors propose to investigate the JKO scheme to approximate the continuous-time Wasserstein gradient flow. Following earlier works on the solution of JKO, they parametrize the solutions of the JKO problems using convex pushforward maps. Using the recently introduced Input Convex Neural Networks (ICCN) and restricting further to the class of pushforward defined by ICNN, the authors obtain an approximate solution of the JKO problem by minimizing the JKO loss w.r.t the parameters of the ICNN. They investigate different energy function (potential, interaction, internal). They then evaluate their methologies on different non-linear PDEs whose solutions or equilibirum are known. Finally, they propose to apply their algorithm for molecular discovery. To do so, they define the JKO problem in the latent space of a VAE.

Main contributions are :
* Extending ICNN + JKO framework to more general energies.
* Application to PDEs + Molecular discovery (comparison with direct optimization).

**Broader Impact Concerns:**

No concern.

**Requested Changes:**

MAJOR :

* More details are needed in the main document, regarding the molecular experiment and in particular how the authors deal with the Sinkhorn or MMD divergence term.

* It should be made more clear that the ICNN + JKO scheme was already introduced in the literature. If I understood correctly the main contribution of the authors in fact resides into the extension of the ICNN + JKO method to other energies.

OTHER COMMENTS :

* Computing and backpropagating through the logarithm of the determinant of the Hessian can be extremely costly as the dimension increase. The authors solve this problem using the Hutchinson method to obtain an unbiased estimator of the trace (which is needed when differenciating). However, more and more samples are needed in order to control the variance of the estimator as the dimension grows. Hence, I am wondering how the method proposed by the authors would really scale to high dimension. Could the authors discuss the limitations of the Hutchinson estimator (if any)?

* In Appendix D, the authors refer to the Stochastic Gradient Langevin Dynamics (SGLD) (there is a typo in the recursion X_K should be X_k). However, the dynamics they use is a very special case of SGLD where there is no noise on the gradient term and therefore SGLD is simply the discretization of the Langevin dynamics (sometimes referred to as ULA - Unadjusted Langevin Algorithm).

* Table 6 is quite hard to read. Highlighting the important and relevant lines would help the reader, trying to check the behavior of the direct optimization baseline.

* Do the authors have any explanation as of why the direct optimization does not work as well as JKO+ICNN? This is an interesting and potentially important findings but it is not obvious to me why JKO would performed better.

* In the molecular experiment the ADAM baselin with \lambda_2 = 0 and 1e-2 give reasonable result except for the final SD (which makes sense since there is no regularization term to fit the distribution to \rho_0). However, would it be possible to compare the point cloud generated with this configuration and the one obtained with JKO?

* In Table 7, the authors compare the speed-up when using the JKO-ICNN to other configurations while the direct approach needs to be retrained entirely. I totally agree that in this case, it will benefit JKO. However (except if I missed something), JKO+ICNN will likely take longer to train in the first place than just the optimization using the direct approach. Could the authors comment on that as well? This will of course depend on the dimensionality + the number of parameters of the ICNN.

**Strengths And Weaknesses:**

STRENGTH:

* The paper is well written with few typos. It is well motivated and the proofs are correct.

* The extension of the parametric JKO setting to internal energies is interesting.

* I also enjoyed the applications to non-linear PDEs and to molecular discovery. In particular, I found interesting the thorough comparison between JKO + ICNN and the "direct approach" (which consists into discretizing the ODE associated with the Wasserstein gradient flow). The JKO approach seems to improve on the direct approach by managing to propose configurations with good QED which are close to the original point cloud.

WEAKNESSES:

* It is hard to understand what are the true contributions of the paper, especially when compared to [1]. In particular, Algorithm 1 was already proposed in [1]. Not taking into the account the other contribution of [1] which considers the learning of the energy, the only difference I can see between the two approaches is that the current paper consider more general energies. From a methodological point of view, this seems incremental. I think the authors should do a better job at motivating the application of ICNN + JKO to these extended energies.

* The molecular discovery application is interesting but not enough details in the main document are given about the functional and its optimization. In particular, how does the divergence term D fits in the general picture presented by the authors? In the supplementary material it is stated that D is either the (entropy regularized) Sinkhorn divergence, or the Maximum Mean Discrepancy. It is not clear to me how such divergence can be understood in the theoretical framework of the authors (even though I agree that the computations can be done by unrolling the minimization in the computation of the divergence).

[1] -- Bunne, Meng-Papaxanthos, Krause, Cuturi -- JKOnet: Proximal optimal transort modeling.

---

> ### Author Response · Authors · 2022-05-31
> **Reviewer x751 Response (3/3)**
>
> **Comparing baseline point cloud to JKO-ICNN point cloud**
> This is an interesting question. When looking at the baseline final point cloud embedding after optimization compared to the final point cloud of embeddings after running JKO-ICNN (for initial sample size of 1,000), we find that there is 17.65 $\pm$ 0.13 (mean $\pm$ one std. come from 5 repeated runs with different seed) Sinkhorn divergence between the two. This is approximately the same Sinkhorn divergence that we found between the baseline and the original point cloud set (see Table 7). This finding is expected, as the JKO-ICNN final point cloud remains quite close to the original.
>
> **JKO-ICNN amortization speed-up compared to initial training time cost**
> The reviewer is correct. Initial training of JKO-ICNN indeed takes a longer time: $745.57 \pm 124.49$ (mean $\pm$ one std. come from 5 repeated runs with different seed) compared to the direct optimization approach: 1.416 seconds.
>
> However, the JKO-ICNN training is a one-time cost that can then be effectively amortized across any new set of points, unlike the direct optimization approach which needs to be incurred for all new sets of embeddings. Therefore by computing the JKO-ICNN maps once offline, our approach offers a much more efficient and flexible solution to direct optimization in an online setting. As detailed in Table 7, the reuse of pre-calculated JKO-ICNN maps is on the order of 10 times faster than solving the direct optimization approach for each new point cloud. Therefore, if the JKO-ICNN maps are re-used for roughly 50-70 point clouds then the computation amortization already pays off.

---

> ### Author Response · Authors · 2022-05-31
> **Reviewer x751 Response (2/3)**
>
> **Explaining poorer performance of the baseline:**
> We found that when optimizing using the baseline approach, i.e. gradient descent on the coordinates of the embeddings, the divergence term dominated the descent path. Therefore, the only baseline hyperparameter configurations that yielded meaningful increases in decoded molecule drug-likeness (QED) values were those where the weight on the divergence term, $\lambda_2$, was orders of magnitude smaller than the one used in the best JKO-ICNN configuration. As a result, the baseline approach was not able to produce final embeddings that stayed close to the original set, an important goal in applications such as drug repurposing.
> It is important to note as well that JKO-ICNN works at the distribution level, and hence in addition to the divergence term and the cost (QED), it uses a Wasserstein proximal method that better constrains the path so that as the cloud point moves it stays close to the “cloud” in the previous step. In contrast, forward methods or gradient descent do  ot have this feature, as they work on particles directly and the only cost available on the distribution level is the divergence term.

---

> ### Author Response · Authors · 2022-05-31
> **Reviewer x751 Response (1/3)**
>
> We thank the reviewer for this thorough review and for the kind words about the quality of our work. Below we address the specific questions and comments:
>
> **Novelty relative to concurrent works:**
> Our work (whose preprint link we do not share here to preserve anonymity) and another one (Mokrov et al., https://arxiv.org/abs/2106.00736) appeared within one hour on the ArXiV. This other work is less general than ours, focusing only on a specific type of functional. The other work (Bunne et al, https://arxiv.org/abs/2106.06345) appeared a few days later on the ArXiv, and although it uses a JKO-ICNN method, it tackles a rather different challenge (the inverse problem), and is thus the least similar of the three.
> Our work did not merely extend an existing JKO-ICNN framework to other functionals not covered in the literature. Rather, we introduced it concurrently to these other works.
>
> **How divergence term fits in our theoretical framework:**
> The divergence term in the molecular discovery experiment stems more from practical needs than theoretical ones. That is, while the divergence term does not correspond to any of the PDEs in Table 1, it is highly relevant for the molecular discovery domain. In the drug repurposing application, for example, it is highly likely that the use case begins with some candidate set of molecules that were verified in a wet lab or by other means that are known to have some desirable properties but not others. In this scenario, one using our framework would want to improve certain properties of the candidate set without deviating too far from it, hence the inclusion of the divergence term in the objective functional.
>
> That said, if the divergence term used was the Wasserstein 2 and not the entropic regularized one, then this cost would also be theoretically motivated since the Wasserstein gradient flow of $W_2$ is related to the dual potential. Moreover, the problem becomes geodesically convex and corresponds to a PDE (Mckean-Vlasov process) with two potentials: 1) the one from KDE and 2) a distribution dependent PDE that is the dual $W_2$ potential. We will clarify this point in our revised submission.
>
>
> **Additional detail in main document regarding the molecular experiment (specifically divergence term)**
> The reviewer’s point is well taken. In our revised work, we will move detail from the Appendix regarding the molecular discovery experiment to the main paper’s Section 6.
>
> **Limitations of the Hutchinson estimator**
> The Hutchinson estimator needs order $\mathcal{O}(1 / \epsilon^2)$ of random projections to give an $(1 \pm \epsilon)$ guarantee for trace approximation. A newer version, Hutch++, was proposed in https://arxiv.org/abs/2010.09649 and reduces the number of projections to $\mathcal{O}(1 / \epsilon)$ random projections. It would be interesting to see how this impacts  the estimation in our case. We will add this discussion to our revised manuscript.
>
> **Typo in Appendix D**
> Thank you for pointing this out. We will fix this in our revised version.
>
> **Clarifying use of SGLD in Appendix D**
> SGLD is the particle implementation corresponding to the PDE with advection (potential energy) and diffusion (entropy). In the particle implementation in SGLD, the potential energy corresponds to the drift and the noise corresponds to the diffusion. SGLD is used to validate JKO-ICNN for advection-diffusion PDE in high dimensions to see if the particle implementation of this PDE and the JKO-ICNN implementation match.
>
> **Cleaning up Table 6**
> We included this table to show that we executed a fair comparison to the direct optimization baseline by executing a significant grid search on the hyperparameters of the baseline approach. However, it is a fair point that this table contains a lot of information and is not straightforward to parse. In our revised work, we will clean this table up visually by breaking it up into sections grouped by hyperparameters and highlighting which configurations performed best in terms of balancing higher decoded QED without sacrificing the validity of the decoded molecules.

---

> > ### Comment · Reviewer_x751 · 2022-06-07
> > **Response**
> >
> > Thank you for your clarifications. I'm quite convinced by the response and the improvements suggested by the authors.
> > I just want to point out that my comment regarding SGLD was maybe misunderstood.
> > In Stochastic Gradient Langevin Dynamics (SGLD) the "Stochastic Gradient" part comes from the fact that we cannot evaluate the drift but we only have access to unbiased samples (compare Equation (4) to Equation (3) in the original SGLD paper). In your work (correct me if I'm wrong) you can compute the drift exactly (i.e. \nabla V) and don't need to have an estimate of the drift. In this case you are using the Langevin Dynamics and not the Stochastic Gradient Langevin Dynamics.

---

### Review · Reviewer_83vF · 2022-05-08

**Summary Of Contributions:**

In this work, authors build on the work of [1] in which it was proposed to solve the JKO scheme by reparametrizing it as a minimization problem over convex functions. In [1], the convex functions were approximated using a space discretization which does not scale in dimension higher than 2. In this paper, it is proposed to use Input Convex Neural Networks (ICNN) to parametrize the set of convex functions. The method is called JKO-ICNN. This has the advantage to scale better with the dimension. Then, experiments are conducted to show that JKO-ICNN are able to learn well Wasserstein gradient flows for simple PDEs. Furthermore, they conduct an experiment in higher dimension in controlled generation for molecular discovery.

[1] Benamou, J. D., Carlier, G., Mérigot, Q., & Oudet, E. (2016). Discretization of functionals involving the Monge–Ampère operator. Numerische mathematik, 134(3), 611-636.

**Broader Impact Concerns:**

No comments for this part.

**Requested Changes:**


I have few remarks on the Related Works section. First, in the paragraph ICNN, optimal transport, and generative modeling, it is written "we consider the setting where the target distribution cannot be sampled from, and is only implicitly characterized as the minimizer of an optimization problem over distributions". I find this sentence confusing since JKO-ICNN can also cover the case where one would want to minimize some loss w.r.t. a distribution from which we have only access to samples. Maybe it should be clarified by saying that the setting here is more general than the other generative modeling works since any loss can basically be plugged in and minimized.

Also, about convex potential flows, it is written that they "are useful for learning generative models when samples from the target (i.e., optimal) distributions are available and the goal is to learn a parametric generative model". While it is true, it can also be used when we only have access to the target up to a constant (using the forward KL instead of the reverse KL).

Another remark is on the related works cited. For example, the JKO scheme (or variations of it) with neural networks is also used in [1, 2, 3]. A method to solve the JKO scheme with the entropic regularization is proposed in [4]. Some works which solve Wasserstein gradient flows using particle schemes (via the forward scheme) might also be cited, e.g. [5,6,7,8,9] for different functionals or [10, 11] for the KL with neural networks.

Typos:
- In Section 4, after quotation of Salim et al, "show that in enjoys"


[1] Lin, A. T., Li, W., Osher, S., & Montúfar, G. (2021, July). Wasserstein proximal of GANs. In International Conference on Geometric Science of Information (pp. 524-533). Springer, Cham.

[2] Fan, J., Taghvaei, A., & Chen, Y. (2021). Variational Wasserstein gradient flow. arXiv preprint arXiv:2112.02424.

[3] Bonet, C., Courty, N., Septier, F., & Drumetz, L. (2021). Sliced-Wasserstein Gradient Flows. arXiv preprint arXiv:2110.10972.

[4] Caluya, K. F., & Halder, A. (2019, July). Proximal recursion for solving the Fokker-Planck equation. In 2019 American Control Conference (ACC) (pp. 4098-4103). IEEE.

[5] Liutkus, A., Simsekli, U., Majewski, S., Durmus, A., & Stöter, F. R. (2019, May). Sliced-Wasserstein flows: Nonparametric generative modeling via optimal transport and diffusions. In International Conference on Machine Learning (pp. 4104-4113). PMLR.

[6] Arbel, M., Korba, A., Salim, A., & Gretton, A. (2019). Maximum mean discrepancy gradient flow. Advances in Neural Information Processing Systems, 32.

[7] Korba, A., Aubin-Frankowski, P. C., Majewski, S., & Ablin, P. (2021, July). Kernel Stein Discrepancy Descent. In International Conference on Machine Learning (pp. 5719-5730). PMLR.

[8] Glaser, P., Arbel, M., & Gretton, A. (2021). KALE Flow: A Relaxed KL Gradient Flow for Probabilities with Disjoint Support. Advances in Neural Information Processing Systems, 34.

[9] Wang, Y., Chen, P., & Li, W. (2021). Projected Wasserstein gradient descent for high-dimensional Bayesian inference. arXiv preprint arXiv:2102.06350.

[10] Feng, X., Gao, Y., Huang, J., Jiao, Y., & Liu, X. (2021). Relative Entropy Gradient Sampler for Unnormalized Distributions. arXiv preprint arXiv:2110.02787.

[11] di Langosco, L. L., Fortuin, V., & Strathmann, H. (2021). Neural Variational Gradient Descent. arXiv preprint arXiv:2107.10731.

**Strengths And Weaknesses:**


Overall, this is a very nice work, well written and with an interesting experiment.

Strengths:
- The paper is well written, and simple to follow
- The molecular discovery is an interesting and original experiment. This shows that gradient flows in high dimension could be used on "real" experiments.

Weaknesses:
- Incremental work (mainly plug ICNN into the JKO scheme, the change of variable formulas are well known for normalizing flows, notably for convex potential flows [1] which are basically the same transformations used in this work, but trained for different objectives).
- High dimensional experiment "only" in R^128
- I believe some points in the experiments could be clarified or improved.

I have some question about the experiments. In Section 5.1, it is stated "here the flow density is estimated from particles via KDE". However, the ICNN learned is supposed to be strictly convex if I understand correctly, and hence we should be able to evaluate directly the density of the flow. Is there a reason why the KDE is used?

In Section 5.2, for the experiment performed, there is no closed-form solution to compare with. But it would be possible for the particular case m=1, a quadratic V and an initial Gaussian. I noticed that it is the setting tested in Appendix D. However, Figure 3 plots the MMD between the JKOICNN trajectory and the Unadjusted Langevin Algorithm (ULA/SGLD). However, in this case, there is a closed-form (see e.g. [2]). Hence, I believe this experiment could be improved. Furthermore, ULA might actually perform worse than JKO-ICNN (see e.g. [2]) so I'm not convinced by this Figure.

For the molecular discovery experiment, I do not believe that the used divergence D used is described in the main part of the paper. I found it in appendix E.2, but I believe it should be in Section 6.

[1] Huang, C. W., Chen, R. T., Tsirigotis, C., & Courville, A. (2020). Convex potential flows: Universal probability distributions with optimal transport and convex optimization. arXiv preprint arXiv:2012.05942.

[2] Mokrov, P., Korotin, A., Li, L., Genevay, A., Solomon, J. M., & Burnaev, E. (2021). Large-scale wasserstein gradient flows. Advances in Neural Information Processing Systems, 34.

---

> ### Author Response · Authors · 2022-05-31
> **Reviewer 83vF Response**
>
> We thank the reviewer for the thoughtful review and for the kind words about the quality of our work. Below we address the reviewer’s specific questions and comments:
>
> **Incremental work:**
> Combining JKO and ICNN is only the beginning of the challenge here. Making this combination practical and implementable for a wide class of functionals potentially in high dimensions requires significant additional effort, tools, and validation, all of which are crucial contributions of our work too, beyond the 'simple' idea of combining JKO and ICNN.
>
> **$\mathbb{R}^{128}$ relatively small:**
> While the high dimensional molecular discovery experiment may seem relatively small compared to, say, 32x32 pixel images, it is still orders of magnitude larger than the dimensionality handled by other PDE solvers. Additionally, other domains with larger dimensionality would also likely go through some initial dimensionality reduction pre-processing to learn a lower dim representation that would likely be of the same order of magnitude as our experiment. Hence, we believe that our molecular discovery experiment indeed constitutes a useful and meaningfully large dimension application.
>
> We also note that at this point, the computational bottleneck is the network size rather than the input dimension (as is the case typically in modern ML), and even so we observe a reasonable dependency on networks’ size. To get a sense of how this dependency could look like, below we present the timing for re-running the molecular discovery experiment but with ICNN hidden dimensionality doubled:
> - Timing for original JKO-ICNN setup presented in Section 6 (i.e., ICNN hidden dim == 100): $745.57 \pm 124.49$ seconds
> - Timing for JKO-ICNN with hidden dim doubled: $816.04 \pm 101.17$ seconds
>
> Mean $\pm$ one std. comes from 5 repeated runs with different seeds. Experiments were run on a V100 GPU. These run times show that our method scales well (sub-linearly) with ICNN capacity.
>
> **Use of KDE:**
> Indeed, the density can be also evaluated from the flow directly. For simplicity, we opted for the KDE estimator. In the final manuscript we will show both KDE and flow estimates.
>
> **Closed-form Solutions for Section 5.2:**
> While the case m=1 does indeed have closed-form solutions and could be used for comparison, this would be an altogether different problem (linear diffusion). Here, we purposely chose the non-linear diffusion case (m>1) precisely because it is substantially more challenging (see Carrillo and Toscani, 2000; Carillo et al. 2021).
>
> **Discussing divergence D in Section 6:**
> The reviewer’s point is well taken. Clarity of this section would be improved if details from Appendix E regarding the divergence D were directly discussed in Section 6. In our revised work, we will be sure to make this change.
>
> **Clarifying the “ICNN, optimal transport, and generative modeling” paragraph:**
> In the case that the reviewer brings up: “minimize some loss w.r.t. a distribution from which we have only access to samples,” while we agree that JKO-ICNN could be used here, the task is different from the ones presented in our work and more direct approaches exist (e.g., generative models, normalizing flows, convex potential flows). However the reviewer’s point is well taken, and we will clarify in our revised work that JKO-ICNN is more general than other generative modeling approaches because, as the reviewer suggests, “any loss can basically be plugged in and minimized.”
>
> **Clarifying the use of convex potential flow:**
> The reviewer raises a good point. We will clarify in our revised submission that convex potential flows can also be used even “when we only have access to the target up to a constant (using the forward KL instead of the reverse KL).”
>
> **Additional references:**
> We thank the reviewer for pointing us to the other relevant and useful works. We will include them as appropriate in our revised submission.
>
> **Typo in Section 4:**
> Thank you for pointing this out. We will fix this in our revised version.

---

### Review · Reviewer_jczu · 2022-05-09

**Summary Of Contributions:**

- The authors propose to use ICNNs to approximate JKO schemes with a number of fixed energy functionals
- The authors validate the proposed method in moderately high dimension (128 dimension VAE latent) and compare to PDEs with known solutions



**Requested Changes:**

I would like the authors' comments on the following points. These are not requests but more request for discussion.

- The authors mention the direct optimization approach multiple times, including in discussion, and use it as the main baseline but do not explain what it is until the appendices. It may be worth commenting what it is briefly in the main text. This is perhaps more taste than needed.

- It is commented briefly on page 10 that direct optimization does not work very well, is there any detailed reason why? The direct optimization approach mentioned in the appendices is interesting as it seems to not require training an ICNN at each iteration. A a side note, this approach could also be used to learn the functional similar to [1] without needing to backprop through the training of an ICNN and hence perhaps more scalable than [1] which appears to use only 4 JKO steps in low dimension.

- It would be useful to give an indication of how long it takes to run a single forward pass of the JKO-ICNN scheme, training ICNN at each step etc., compared to the other approaches and detail the effect of warm-start as it is not clear if it is used.

- It appears the ICNNs used are relatively small, just two layers with width 100. Is this standard in ICNN literature or do practitioners typically use larger networks? Does this impact the training time and feasibility of the proposed method for a large number of JKO steps?

- The experiment runs at dimension 128 in the latent space of a VAE, does the method scale to higher dimensions or will this require larger ICNNs and hence longer training per time-step and would this be prohibitively slow?

[1] Proximal Optimal Transport Modeling of Population Dynamics, Bunne 2021

**Strengths And Weaknesses:**

- Strengths
  - Using ICNNs to approximate JKO schemes appears to be a useful methodological contribution and it appears to perform well empirically and scales to moderately high dimension
  - Experiments are interesting, thorough, and validate the method both for closed-form PDEs and for the molecular experiment at dimension 128 (latent of VAE)

- Weaknesses
  - My understanding is that ICNNS require training at each time-step or fine-tuning from a warm-start initialization. I feel this would be quite slow for a large number of JKO steps if training to completion.
  - On page 9 it is mentioned that the authors do not use warm-start, use only 500 inner training steps for the ICNN, and a relatively small only 2 layer ICNN, does this converge in so few steps? Does warm-start not work given it is not used? I imagine pretraining without meta-learning may result in the network being initialized from a local mode and not adapting.

  - Although TMLR does not review for novelty or originality and I agree that it may be concurrent, there is little methodological contribution beyond previously published [1].  I understand that unlike [1], the authors demonstrate the proposed method in high dimensional settings, for a larger number of JKO steps, however the underlying methodological contribution is very similar. It may be worth expanding on the differences.


[1] Proximal Optimal Transport Modeling of Population Dynamics, Bunne 2021

---

> ### Author Response · Authors · 2022-05-31
> **Reviewer jczu Response (2/2):**
>
> **Size of ICNN relative to input dimension and effect on training time**
> As the reviewer suggests, for higher dimensional problems the capacity of the ICNN will likely need to be increased. However, by demonstrating our JKO-ICNN approach on a 128-dimensional problem (which is orders of magnitude higher than the dimensionality handled by other PDE solvers) we have shown that the approach itself is scalable.
> In terms of getting a sense of how this scaling could look like, below we present the timing for re-running the molecular discovery experiment but with ICNN hidden dimensionality doubled:
> - Timing for original JKO-ICNN setup presented in Section 6 (number of original embeddings $N$ set 1,000, number of iterations for inner loop $n_u$ set to 500, number of JKO steps set to 100, ICNN hidden dim set to 100):
> $745.57 \pm 124.49$ seconds
> - Timing for JKO-ICNN with hidden dim doubled (ICNN hidden dim == 200): $816.04 \pm 101.17$ seconds
>
> Mean $\pm$ one std. comes from 5 repeated runs with different seeds. Experiments were run on V100 GPU. These run times show that our method scales well (sub-linearly) with ICNN capacity.

---

> ### Author Response · Authors · 2022-05-31
> **Reviewer jczu Response (1/2):**
>
> We thank the reviewer for the useful and in-depth feedback on our work and for recognizing the usefulness in higher dimensions and more impactful applications, such as molecular discovery. Below we address the reviewer’s concerns and questions:
>
> **Computational cost of training ICNNs at each JKO step:**
> While our method indeed requires training an ICNN at each JKO step, we found that this was not prohibitively expensive. As detailed below, even in the high dimensional case of the molecular discovery, a single JKO step takes about 6-9 seconds (for a batch size of 1000) and the entire flow (e.g., 100 JKO steps) completes in under 20 minutes when running on a single V100 GPU (see comments below for more details).
>
> **Use of warmstart:**
> We found through experimentation that while warm-starting was helpful in speeding up convergence in the lower dimensional cases, it yielded poorer performance in the high dimensional molecular discovery application, in terms of moving the latent embeddings to spaces that would decode to higher drug-likeness (QED) values.
> In our revised work, we will clarify that warmstart **was used** in solving PDEs with known solutions (Section 5) and was **not** used in the molecular discovery experiments (Section 6).
>
> **Novelty relative to concurrent work:**
> Our work (whose preprint link we do not share here to preserve anonymity) and another one (Mokrov et al., https://arxiv.org/abs/2106.00736) appeared within one hour on the ArXiV. This other work is less general than ours, focusing only on a specific type of functional. The other work (Bunne et al, https://arxiv.org/abs/2106.06345) appeared a few days later on the ArXiv,, and although it uses a JKO-ICNN method, it tackles a rather different challenge (the inverse problem), and is thus the least similar of the three.
>
> **Explaining the baseline in the main text**
> The reviewer’s point is well taken. In our revised submission, we will move more of the text explaining the direct optimization baseline, specifically its formal definition, from the appendix to the main paper’s experimental results sections.
>
> **Why the baseline performs more poorly on joint objective**
> We found that when optimizing using the baseline approach, i.e. gradient descent on the coordinates of the embeddings, the divergence term dominated the descent path. Therefore, the only baseline hyperparameter configurations that yielded meaningful increases in decoded molecule drug-likeness (QED) values were those where the weight on the divergence term, $\lambda_2$, was orders of magnitude smaller than the one used in the best JKO-ICNN configuration. As a result, the baseline approach was not able to produce final embeddings that stayed close to the original set, an important goal in applications such as drug repurposing.
> It is important to note that JKO-ICNN works at the distribution level, and hence in addition to the divergence term and the cost (QED), it uses a Wasserstein proximal method that better constrains the path so that as the cloud point moves, it stays close to the “cloud” in the previous step. In contrast, forward methods or gradient descent do not have this feature, as they work on particles directly and the only cost available at the distribution level is the divergence term.
>
>
> **Using the baseline to learn the functional:**
> This is indeed discussed in Bunne et al as well. However, as the authors discuss there, this approach (which they call the “Forward method”) is
> 1. not as robust to noise as using JKO with ICNNs (see Figure 5 in Bunne et al) and
> 2. does not extrapolate to unseen data (see Figures 10 and 11 in Bunne et al).
>
> Indeed this second point about extrapolation mirrors the argument we make as to why our approach is better than the baseline. While the baseline can optimize for a given set of embeddings, any new point cloud would need to be trained from scratch (e.g., no extrapolation as argued in Bunne et al). The JKO-ICNN approach however can be re-used to transport any arbitrary new point cloud, and does so well (see Table 7 in our work). Thus the initial computational cost of training JKO-ICNN on one point cloud can be effectively amortized on all new sets of embeddings.
>
> **Timing of one JKO-ICNN step**
> For the molecular discovery experiment described in Section 6, i.e., number of original embeddings $N$ set 1,000, JKO rate $\tau$ set to 1e−4, number of iterations for inner loop $n_u$ set to 500, and inner loop learning rate $\eta$ set to 1e−3, we find that one JKO step (i.e., 500 inner loop steps) takes about 6-9 seconds (on a V100 GPU).

---

### Author Response · Authors · 2022-07-05
**Revised manuscript posted**

Dear Editor and Reviewers,

We posted a revised manuscript that addresses reviewer questions and clarification requests. Please see the comment accompanying the revision above for details of changes.

Thank you again for your kind feedback and for overseeing our submission.


Best,

Authors

---

### Comment · Action_Editors · 2022-07-11
**Accept with minor revisions**

The paper has now been reviewed by three reviewers: two recommend accept and one leans towards acceptance (however one of the main concerns of this reviewer is novelty but, as pointed out by the authors, the first version of this work appeared at the same time as other JKO-neural net schemes).
Thus I recommend acceptance of the paper.

A minor point has been raised by reviewer x751: Appendix D appears to use erroneously the terminology SGLD. SGLD is an approximation of Unadjusted Langevin algorithm (ULA) using a Monte Carlo approximation of the gradient (mini batches in the original paper of Welling & Teh). It seems you are using here ULA and not SGLD. Please correct it.

---

> ### Author Response · Authors · 2022-07-11
> **Thank you, Revision uploaded**
>
> Dear Action editor,
>
> Thank you very much for overseeing our paper and for the constructive feedback. We have revised Appendix D according to your recommendation and uploaded it to the portal. We changed SGLD throughout to ULA.
>
> Thank you again.
>
>
> Best regards,
>
> Authors

---

### Decision · Action_Editors · 2022-07-14

**Recommendation:** Accept with minor revision

**Comment:**

The paper makes an interesting contribution by developing a novel scheme to implement gradient flows optimizing functionals on the space of probability distributions. It proposes an approximation of the celebrated JKO scheme using ICNN. The methodology is demonstrated on a variety of applications including molecular discovery.

The paper has now been reviewed by three reviewers: two recommend accept and one leans towards acceptance (however one of the main concerns of this reviewer is novelty but, as pointed out by the authors, the first version of this work appeared at the same time as other JKO-neural net schemes). Thus I recommend acceptance of the paper.

A minor point has been raised by reviewer x751: Appendix D appears to use erroneously the terminology SGLD. SGLD is an approximation of Unadjusted Langevin algorithm (ULA) using a Monte Carlo approximation of the gradient (mini batches in the original paper of Welling & Teh). It seems you are using here ULA and not SGLD. Please correct it.